# Agent-GWO: Collaborative Agents for Dynamic Prompt Optimization in Large Language Models

## Abstract

Large Language Models (LLMs) have demonstrated remarkable capabilities in natural language understanding and generation, yet their performance in complex reasoning tasks remains limited. A central challenge lies in their heavy reliance on manually designed static prompts, which are costly to engineer, lack flexibility, and often fail to generalize across diverse tasks. In this work, we propose Agent-GWO, a dynamic prompt optimization framework that leverages collaboration among multiple LLM-based agents and the Grey Wolf Optimizer (GWO). Instead of fine-tuning model parameters, Agent-GWO enhances reasoning by iteratively refining task-specific prompts through cooperative optimization. Each agent is modeled as a "wolf," guided by its hyperparameters and reasoning template. Through GWO's hierarchical leader–follower mechanism, top-performing leader agents ($\alpha$, $\beta$, and $\delta$) guide the evolution of other agents, enabling the population to converge toward robust and effective reasoning strategies. Extensive experiments across mathematical reasoning, hybrid reasoning, and domain-specific applications (e.g., social sciences, medical diagnostics, and decision support) demonstrate the effectiveness of our approach. For example, on GPT-4.1-mini, Agent-GWO improves GSM8K accuracy by 8.7% (from 88.2% to 96.9%) and MMLU accuracy by 12.9% (from 66.9% to 79.8%).

## 1 Introduction

In recent years, LLMs (Vaswani et al., 2017; Devlin et al., 2019; Brown et al., 2020; Hadi et al., 2023; Achiam et al., 2023) have achieved remarkable advancements in natural language understanding and generation. These models demonstrate broad application potential across diverse domains, including social sciences, humanities, healthcare, and business decision-making (Bommasani et al., 2021; Singhal et al., 2023; Kasneci et al., 2023; Chui et al., 2023). While mainstream LLMs excel in complex tasks such as task decomposition, behavior planning, and code generation, they encounter significant challenges in reasoning-based tasks. These challenges include the lack of effective analytical methods for domain-specific knowledge and poor logical consistency in complex reasoning scenarios (Raffel et al., 2020; Bubeck et al., 2023; Lu et al., 2023; Hadi et al., 2023).

To enhance the reasoning capabilities of LLMs in complex tasks, researchers have pursued multidimensional explorations in model architecture, data scale, and reasoning strategies. At the architectural level, scaling model size and incorporating sophisticated attention mechanisms, such as variants of the Transformer (Vaswani et al., 2017), have bolstered semantic representation capabilities. On the data front, pre-training on large-scale multimodal datasets and fine-tuning with domain-specific corpora have significantly improved model performance in specialized tasks (Bommasani et al., 2021). During inference, Chain-of-Thought (CoT) reasoning and self-correction mechanisms improve reasoning accuracy in complex scenarios (Wei et al., 2022). However, it is essential to acknowledge that LLMs still encounter substantial challenges in complex reasoning tasks, which often necessitate multi-step logical analysis and the integration of diverse information, such as solving intricate mathematical problems or handling ambiguous scenarios. Furthermore, approaches such as CoT and self-refinement strategies rely heavily on manually crafted prompts, lacking robustness and frequently leading to logical errors or hallucinations (Wei et al., 2022; Zhang et al., 2025). In addition, scaling up model

size to improve performance results in increased computational costs, limiting the applicability of LLMs in resource-constrained environments (Dodge et al., 2020; Zheng et al., 2025).

Inspired by swarm intelligence optimization, we propose a novel LLM-driven multi-agent collaborative optimization framework based on the Grey Wolf Optimizer (GWO) (Mirjalili et al., 2014) to optimize a population of LLM agents iteratively, thereby enhancing their reasoning capabilities in complex tasks. GWO demonstrates strong performance through its simplicity, rapid convergence, and minimal hyperparameter requirements, enabling its seamless integration into various optimization processes, such as reasoning chain design, behavior strategy planning, and hyperparameter selection.

As illustrated in Figure 1, our framework abstracts each LLM agent as a "wolf" within a population, leveraging GWO's hierarchical hunting mechanism to introduce a layered collaborative optimization strategy. Candidate solutions are defined by the agent's hyperparameter configuration and reasoning prompt template. The algorithm begins with a diverse initialization of the agent population. In each optimization iteration, generative agents execute CoT reasoning tasks based on their configurations, while evaluative agents quantitatively score their outputs across three dimensions: logical consistency, creativity, and reasoning completeness. Based on fitness rankings, the top three leader agents—denoted as $\alpha$, $\beta$, and $\delta$—are dynamically selected. The remaining agents, guided by GWO's hierarchical update mechanism, adaptively optimize by incorporating features from the leaders.

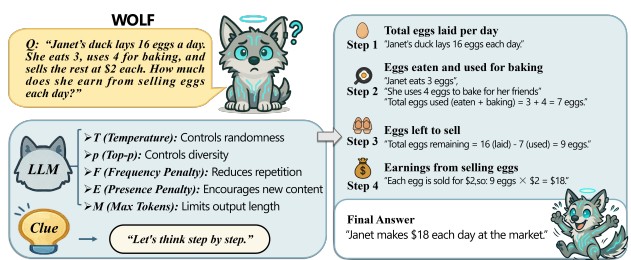

Figure 1: GWO abstracts each LLM agent as a "wolf" which is characterized by hyperparameters & reasoning prompt. These two guide the behavior of the LLMs during the optimization process.

This collaborative process iterates over multiple rounds, effectively enabling a global search of reasoning strategies and parameter spaces, ultimately converging to the optimal agent configuration. This approach significantly enhances LLM reasoning performance in complex task scenarios. To validate the effectiveness of our framework, we conduct extensive experiments on various tasks, including social science question answering, medical diagnostics, and business decision support.

Our main contributions are threefold:

- We find that dynamically optimizing prompts during training enables LLMs to develop more effective, task-specific strategies. Based on this, we propose a parameter-agnostic adaptation paradigm that avoids fine-tuning or adding components, thereby reducing both training and inference costs.
- By combining multi-agent collaboration with the hierarchical optimization of the Grey Wolf Optimizer, we introduce an iterative leader–agent optimization mechanism to enhance the robustness and predictability of multi-agent collaboration.
- Extensive experiments show that our method consistently surpasses strong baselines, delivering superior performance with low computational overhead.

## 2 RELATED WORK

### 2.1 PROMPT ENGINEERING AND STATIC CHAIN-OF-THOUGHT METHODS

Static prompting remains a foundational technique for adapting LLMs in zero- and few-shot scenarios, though it relies heavily on manually crafted exemplars (Brown et al., 2020). Chain-of-Thought (CoT) prompting (Wei et al., 2022) introduced stepwise reasoning demonstrations, inspiring variants such as self-consistency (Wang et al., 2022a) and least-to-most decomposition (Zhou et al., 2022). Other extensions include Self-Ask with Search (Press et al., 2022), Program-of-Thoughts (PoT), which generates executable code traces (Chen et al., 2022), and Graph-of-Thoughts (GoT), which models reasoning as dependency graphs (Liu et al., 2023).

Despite these advances, static methods face critical limitations: they require human-designed exemplars for each task (Zhang et al., 2022), remain highly sensitive to prompt order and phrasing (Lu et al., 2022), and incur high inference costs due to lengthy reasoning chains (Zhou et al., 2022). More importantly, once exemplars and templates are fixed, the reasoning process becomes static, making it difficult to adapt dynamically to new contexts or evolving tasks.

## 2.2 Collaborative Reasoning with Multi-Agent LLMs

Beyond single-agent prompting, researchers have explored collaborative reasoning via Multi-Agent Systems (MAS), which originate from Distributed Artificial Intelligence and emphasize autonomy, cooperation, and coordination (Wooldridge & Jennings, 1995; Ferber & Weiss, 1999). The integration of LLMs (Naveed et al., 2023; Kumar, 2024) into MAS has significantly expanded their capacity in natural language-based planning, programming, and reasoning (Xi et al., 2025). Representative frameworks such as AutoGen (Wu et al., 2023) and Chain of Agents (Zhang et al., 2024b) exemplify this trend by enabling dynamic task decomposition and agent collaboration.

Recent advances include modular LLM-agent architectures (Liu et al., 2025), multi-round debates for collective reasoning (Du et al., 2023), and communication protocols based on DCOP (Fioretto et al., 2018), FIPA-ACL (Fipa, 2002), or attention mechanisms (Jiang & Lu, 2018). Applications span social simulation in Smallville (Park et al., 2023), recommender systems (Zhang et al., 2024a), intelligent manufacturing (Lim et al., 2024), and macroeconomic modeling (Li et al., 2023).

However, existing multi-agent reasoning approaches typically follow a one-off collaboration paradigm, where agents jointly generate solutions but lack iterative refinement mechanisms. In contrast, our work is the first to combine multi-agent collaboration with an iterative leader–agent refinement mechanism, where leaders guide and update agents across multiple rounds. This iterative paradigm enables dynamic optimization of reasoning paths and collaboration strategies, distinguishing our framework from both static prompting methods (CoT, ToT, GoT) and conventional MAS-based reasoning.

## 2.3 Swarm Intelligence Optimization

Swarm Intelligence Algorithms (SIA) address complex optimization by simulating collective behaviors Chakraborty & Kar (2017). Foundational methods include Particle Swarm Optimization (PSO) Kennedy & Eberhart (1995) and Ant Colony Optimization (ACO) Dorigo (1992), with recent extensions integrating SIA into DL and LLMs Wang et al. (2018); Shriyan et al.. Among them, Grey Wolf Optimizer (GWO) (Mirjalili et al., 2014) is notable for efficiency and simplicity, with enhanced variants addressing complex search spaces Meidani et al. (2022); Zhang et al. (2021). GWO has been applied in engineering, NLP, and vision tasks, and in our work, we extend its hierarchical optimization to multi-agent collaboration, enabling iterative refinement in LLM-based reasoning.

For a comprehensive review, please refer to Appendix A.1.

## 3 Method

### 3.1 Theoretical Background and Problem Formulation

**Grey Wolf Optimizer**. GWO is a population-based metaheuristic inspired by the social hierarchy and hunting behavior of grey wolves. Designed for solving continuous optimization problems, GWO is characterized by its simplicity and minimal reliance on hyperparameters. In this algorithm, a population of $N$ wolves is maintained, where each wolf represents a candidate solution $\mathbf{X}_i = (x_i^1, x_i^2, \ldots, x_i^D)$ in a $D$-dimensional search space. The objective is to optimize a target function $f(\mathbf{X})$. GWO mimics the hierarchical structure of grey wolves, dividing the population into four roles: $\alpha$ (leader), $\beta$ (second-in-command), $\delta$ (subordinate), and $\omega$ (follower). During the optimization process, the $\alpha$, $\beta$, and $\delta$ wolves correspond to the top three solutions and are responsible for guiding the remaining $\omega$ wolves toward promising regions in the search space.

The core mechanism of the GWO is inspired by the encircling behavior exhibited during hunting. In each iteration, wolves estimate their distance from the prey (i.e., the current best solution) as $\mathbf{D}(t) =$

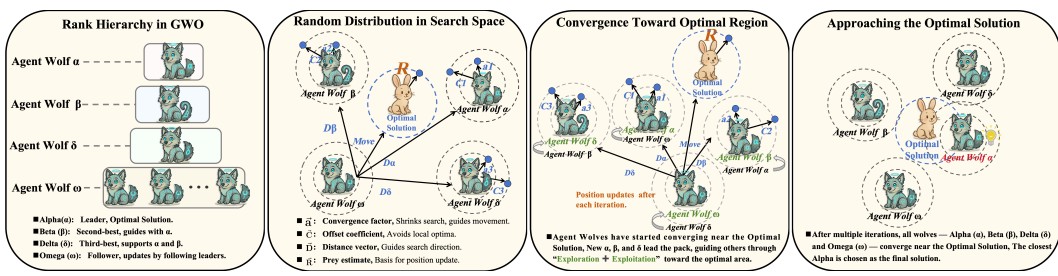

Figure 2: Illustration of original GWO algorithm.

$|\mathbf{C}(t) \odot \mathbf{X}_p(t) - \mathbf{X}_i(t)|$, where $\mathbf{X}_p(t)$ denotes the position of the prey at iteration $t$, $\mathbf{C}(t) = 2\mathbf{r}_1(t)$, and $\mathbf{r}_1(t) \in [0,1]^D$ is a uniformly distributed random vector.

Based on $\mathbf{D}(t)$, the position of each wolf is updated by $\mathbf{X}_i(t+1) = \mathbf{X}_p(t) - \mathbf{A}(t) \odot \mathbf{D}(t)$, where $\mathbf{A}(t) = 2\mathbf{a}(t) \odot \mathbf{r}_2(t) - \mathbf{a}(t)$ and $\mathbf{r}_2(t) \in [0,1]^D$ is another random vector. The parameter decreases linearly with iterations as $\mathbf{a}(t) = 2 - 2 \cdot \frac{t}{T_{\max}}$, where $T_{\max}$ is the maximum number of iterations. This design allows the algorithm to balance exploration in early stages and exploitation in later stages.

Each $\omega$ wolf also updates its position by referring to the three leading wolves $\alpha$, $\beta$, and $\delta$. The distances are computed as $\mathbf{D}_j(t) = |\mathbf{C}_j(t) \odot \mathbf{X}_j(t) - \mathbf{X}_i(t)|$, $j \in \{\alpha, \beta, \delta\}$, and the corresponding candidate updates are $\mathbf{X}_i^{(j)}(t+1) = \mathbf{X}_j(t) - \mathbf{A}_j(t) \odot \mathbf{D}_j(t)$. Finally, the new position of wolf $i$ is obtained by averaging the three guided positions:

$$\mathbf{X}_i(t+1) = \frac{\mathbf{X}_i^{(\alpha)}(t+1) + \mathbf{X}_i^{(\beta)}(t+1) + \mathbf{X}_i^{(\delta)}(t+1)}{3} \tag{1}$$

By integrating leader-based guidance with stochastic exploration, the GWO effectively achieves a balance between global search and local exploitation. As iterations progress, the decreasing parameter $\mathbf{a}(t)$ ensures a smooth transition from exploration to exploitation, enhancing convergence toward the global optimum.

**Agent Structure and Definition**. Suppose there are $n$ agents. The $j$-th agent, denoted as $\text{Agent}_j$, consists of a large language model $\text{LLM}_j$ and a prompt template $\text{prompt}_j$, i.e., $\text{Agent}_j = \{\text{LLM}_j, \text{prompt}_j\}$. The language model $\text{LLM}_j$ contains a shared model parameter set $\boldsymbol{\theta}$ and an agent-specific hyperparameter set $\boldsymbol{\eta}_j = \{T_j, p_j, F_j, E_j, M_j\}$, which represent temperature, top-$p$ threshold, frequency penalty, presence penalty, and maximum token length, respectively. Therefore, $\text{LLM}_j = \{\boldsymbol{\eta}_j, \boldsymbol{\theta}\}$, and $\text{Agent}_j = \{\{\boldsymbol{\eta}_j, \boldsymbol{\theta}\}, \text{prompt}_j\}$.

**Hyperparameter Sampling.** To ensure diversity and stability in agent behavior, we design a hyperparameter sampling strategy for the set $\boldsymbol{\eta}_j = \{t_j, p_j, f_j, e_j, m_j\}$, where a clipping function $\text{clip}(x, [a, b]) = \max(a, \min(x, b))$ constrains sampled values within valid ranges, thereby maintaining controlled and consistent generation. Specifically, the temperature $t_j$ and the top-$p$ threshold $p_j$ are independently drawn from normal distributions $\mathcal{N}(\mu_t, \sigma_t^2)$ and $\mathcal{N}(\mu_p, \sigma_p^2)$, then clipped to intervals $[a_t, b_t]$ and $[a_p, b_p]$, respectively. These parameters regulate the generation distribution: higher values enhance creativity and diversity, while lower values promote predictability and stability. The frequency penalty $f_j$, sampled from $\mathcal{N}(\mu_f, \sigma_f^2)$ and clipped to $[a_f, b_f]$, mitigates repetitiveness by penalizing tokens proportionally to their prior frequency. The presence penalty $e_j$, sampled from $\mathcal{N}(\mu_e, \sigma_e^2)$ and clipped to $[a_e, b_e]$, encourages novelty by uniformly penalizing any previously generated token, thus fostering the introduction of new words or concepts. Finally, the maximum token length $m_j$ is set to a fixed constant $c_m$ for tasks requiring predetermined length, while for adaptive-length tasks it is uniformly drawn from a discrete set $\mathcal{M}$, i.e., $m_j \sim \text{Uniform}(\mathcal{M})$.

This sampling approach uses normal or uniform distributions to introduce controlled variability, while clipping ensures hyperparameters remain within reasonable bounds, optimizing creativity, stability, and content quality.

**Problem Formulation.** We denote the reasoning task dataset as $\mathcal{D} = \{q_1, q_2, \ldots, q_N\}$, where each element $q_i$ represents a specific question. The dataset consists of $N$ question samples in total. During

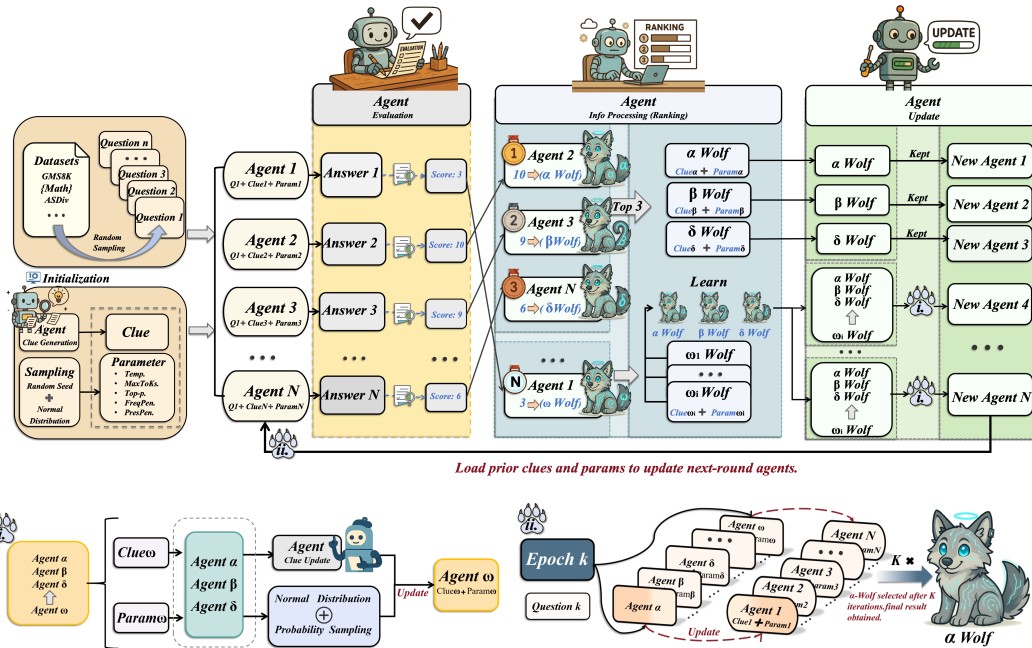

Figure 3: The overall GWO framework operates by having each agent process a dataset, clue, and parameters to produce outputs, which are then scored. Only the top-3 agents ($\alpha$, $\beta$, and $\delta$) are differentiated, while the rest (i.e., $\omega$) are updated using the clue and parameters from the previous round. This iterative ranking and update process continues until termination, after which the final agent is set as the top-ranked $\alpha$ from the last iteration. **Note:** The left-bottom subfigure illustrates the learning process of each agent at each iteration and the right-bottom subfigure illustrates the optimization process.

the reasoning process, given any question $q \in \mathcal{D}$, a specific agent, denoted as $\text{Agent}_j$, processes the question and outputs two components: first, a detailed chain of thought (denoted as $\text{CoT}_j$), which illustrates the step-by-step reasoning process taken by the agent; and second, a final answer (denoted as $\text{Answer}_j$) that the agent derives based on the chain of thought. This reasoning process can be formally expressed as a function:

$$f(\text{Agent}_j, q) = (\text{CoT}_j, \text{Answer}_j), \quad q \in \mathcal{D} \tag{2}$$

## 3.2 MULTI-AGENT GREY WOLF OPTIMIZER FRAMEWORK

The **GWO algorithm**, outlined in Algorithm 2 and illustrated in Figure 3, adapts GWO to optimize LLM configurations by modeling agents as wolves in a population. Each agent ($\text{Agent}_j = (\boldsymbol{\eta}_j, \text{prompt}_j)$) represents a candidate solution with a unique set of hyperparameters and a prompt template. Figure 2 elucidates the methodology for addressing continuous optimization challenges through the emulation of grey wolves' social hierarchy and hunting behaviors.

**Initialization.** The algorithm initializes a population $\mathcal{A}$ of n agents. For each agent $\text{Agent}_j$, hyperparameters $\boldsymbol{\eta}_j$ are randomly sampled from predefined ranges (e.g., temperature $\in [0, 1]$, top-p $\in [0, 1]$) using a normal distribution. Similarly, the prompt $\text{prompt}_j$ is sampled from a set of predefined CoT prompt templates or generated via a template generation function. This diverse initialization ensures broad exploration of the search space.

**Optimization Loop.** The optimization phase iterates $K$ times, with each iteration evaluating and updating the agent population. For each agent $Agent_j$, a question $q$ is sampled from $\mathcal{D}$, and the LLM generates a CoT trace and answer using the agent's configuration. The fitness of the answer is computed using a carefully designed evaluation function, which assesses the response based on three

crucial dimensions: the logical consistency of the reasoning process, the ingenuity of the reasoning approach, and the comprehensiveness of the reasoning content. This multi-dimensional evaluation ensures a thorough and accurate assessment of the answer's quality. The top three agents, denoted $\alpha$, $\beta$, and $\delta$, are selected based on their fitness scores, representing the best, second-best, and third-best solutions, respectively.

Non-elite agents ($\mathcal{A} \setminus \{\alpha, \beta, \delta\}$) are updated to converge toward the top performers. For each hyperparameter $\eta_j^{(k)}$ in agent $Agent_j$, new values are sampled from normal distributions centered at the top agents: $X_r \sim \mathcal{N}(\eta_r^{(k)}, \sigma^2)$, for $r \in \{\alpha, \beta, \delta\}$, and combined as a weighted average: $\eta_j^{(k)} = w_\alpha X_\alpha + w_\beta X_\beta + w_\delta X_\delta$, with weights satisfying $w_\alpha > w_\beta > w_\delta$ and $w_\alpha + w_\beta + w_\delta = 1$. This prioritizes the influence of the best- performing agent ($\alpha$) while maintaining diversity. The prompt is adapted by a function PROMPTADAPTATION($\cdot$), which blends features of the top prompts (e.g., via template mixing or keyword imitation). All hyperparameters are clipped to remain within valid bounds. After $K$ iterations, the configuration of the $\alpha$ agent, ($\boldsymbol{\eta}_\alpha$, $prompt_\alpha$), is returned as the optimal solution ($\boldsymbol{\eta}^*$, prompt$^*$).

**Overall framework.** The integration of the GWO with LLMs in our framework is achieved through the coordinated interaction of four specialized agents. The process begins with GENERATION($\cdot$), which queries the LLM using specific configurations composed of hyperparameters ($\boldsymbol{\eta}_j$) and prompt templates (prompt$_j$). These agents are responsible for producing step-by-step CoT reasoning traces and final answers to the given questions. Once the responses are generated, they are passed to EVALUATION($\cdot$), which assesses the quality of the CoT and answers across three dimensions: logical consistency, creativity, and completeness. The resulting fitness scores are then collected and processed by RANKING($\cdot$), which sorts all candidate solutions and identifies the top three performing agents—$\alpha$, $\beta$, and $\delta$—to serve as leaders in the GWO hierarchy. These rankings guide the subsequent optimization phase. Finally, UPDATE($\cdot$) utilizes the evaluation feedback and ranking outcomes to adjust the prompt templates and potentially refine the hyperparameters for the next generation. This iterative process continues over multiple rounds, enabling the system to converge toward more effective prompt generation strategies and reasoning behaviors, while maintaining the core dynamics of the GWO algorithm within the LLM setting.

## 4 EXPERIMENTS

In this section, we conduct a systematic evaluation of the proposed GWO-based multi-agent collaboration framework to verify its effectiveness on complex reasoning tasks. Our evaluation covers the reasoning performance (Sec 4.2), adaptability performance to CoT (Sec 4.3), and ablation study (Sec 4.4).

### 4.1 EXPERIMENTAL SETUP

To evaluate the proposed GWO framework on complex reasoning tasks, we use datasets spanning two categories: mathematical reasoning (GSM8K, MATH, SVAMP, MULTIARITH, ASDIV, MATH_MIX) and hybrid reasoning (AQUA, MMLU, BBH, DATE, CLUTRR). These benchmarks cover arithmetic, algebra, geometry, logic, and interdisciplinary reasoning, testing both numerical computation and qualitative inference. We assess performance across diverse models: Qwen2.5-Coder-7B-Instruct, optimized for code and math; GPT-4o-mini, a lightweight multimodal model; GPT-4.1-mini and GPT-4.1-nano, efficiency-oriented GPT-4.1 variants; and Gemma-3-12b-it, a 12B instruction-tuned model strong in math and logic. This diversity enables robust evaluation under heterogeneous settings. Unless noted otherwise, experiments use the default GWO setup with $n = 5$ agents and $K = 10$ iterations.

### 4.2 PERFORMANCE EVALUATION

To evaluate the effectiveness of the proposed GWO framework, we conducted extensive experiments on several mainstream LLMs across diverse benchmarks, including mathematical and hybrid reasoning tasks. The results, as shown in Tables 1 and 2, demonstrate that the GWO framework significantly improves reasoning accuracy across different models.

Table 1: Performance evaluation of GWO on math reasoning task.

| Model | Math Reasoning Tasks | | | | | |
|---|---|---|---|---|---|---|
| | GSM8K | MATH | SVAMP | MultiArith | ASDiv | AQUA |
| GPT-4o-mini | | | | | | |
| CoT | 85.3% | 78.1% | 83.9% | 98.7% | 91.1% | 65.3% |
| CoT-SC/n=5 | 90.2% | 78.6% | 85.8% | 99.1% | 92.8% | 70.1% |
| GWO | **94.9%** | **79.2%** | **92.3%** | **99.3%** | **94.5%** | **75.9%** |
| GPT-4.1-mini | | | | | | |
| CoT | 88.2% | 79.8% | 86.1% | 99.0% | 91.7% | 67.5% |
| CoT-SC/n=5 | 91.8% | 83.2% | 87.9% | 99.4% | 93.2% | 71.9% |
| GWO | **96.9%** | **83.4%** | **92.7%** | **99.6%** | **94.8%** | **78.5%** |
| GPT-4.1-nano | | | | | | |
| CoT | 83.8% | 74.3% | 81.1% | 98.8% | 89.6% | 64.7% |
| CoT-SC/n=5 | 87.7% | 77.4% | 83.1% | 99.2% | 91.2% | 68.5% |
| GWO | **92.5%** | **79.4%** | **91.8%** | **99.3%** | **93.1%** | **76.4%** |
| Qwen2.5-Coder-7B-Instruct | | | | | | |
| CoT | 77.3% | 69.7% | 82.1% | 92.1% | 86.4% | 60.4% |
| CoT-SC/n=5 | 80.1% | 71.2% | 84.2% | 94.9% | 88.7% | 62.1% |
| GWO | **89.1%** | **72.1%** | **90.1%** | **97.1%** | **91.2%** | **62.3%** |
| Gemma-3-12b-it | | | | | | |
| CoT | 83.5% | 76.5% | 79.3% | 90.2% | 88.1% | 69.1% |
| CoT-SC/n=5 | 85.8% | 78.7% | 81.1% | 93.3% | 90.3% | 71.4% |
| GWO | **92.8%** | **80.1%** | **90.9%** | **95.9%** | **93.7%** | **78.5%** |

Table 2: Performance evaluation of GWO on hybrid reasoning datasets.

| Model | General | Multitask | Temporal | Logical | Math Mix |
|---|---|---|---|---|---|
| | MMLU | BBH | Date | CLUTRR | MATH_MIX |
| GPT-4o-mini | | | | | |
| CoT | 62.8% | 66.3% | 51.7% | 66.1% | 82.5% |
| CoT-SC/n=5 | 67.6% | 68.9% | 54.7% | 72.6% | 85.0% |
| GWO | **73.3%** | **70.9%** | **76.3%** | **74.4%** | **87.5%** |
| GPT-4.1-mini | | | | | |
| CoT | 66.9% | 69.5% | 54.9% | 71.2% | 82.5% |
| CoT-SC/n=5 | 71.2% | 72.6% | 57.9% | 76.5% | 85.0% |
| GWO | **78.3%** | **76.7%** | **78.7%** | **78.6%** | **92.0%** |
| GPT-4.1-nano | | | | | |
| CoT | 61.3% | 65.7% | 52.1% | 64.2% | 80.0% |
| CoT-SC/n=5 | 65.8% | 68.2% | 55.6% | 69.3% | 83.0% |
| GWO | **73.5%** | **69.1%** | **77.1%** | **71.6%** | **88.0%** |
| Qwen2.5-Coder-7B-Instruct | | | | | |
| CoT | 55.1% | 47.2% | 31.1% | 20.2% | 73.5% |
| CoT-SC/n=5 | 56.1% | 49.5% | 32.9% | 21.1% | 75.5% |
| GWO | **58.3%** | **53.9%** | **37.1%** | **27.8%** | **83.5%** |
| Gemma-3-12b-it | | | | | |
| CoT | 68.3% | 64.1% | 77.9% | 49.3% | 78.0% |
| CoT-SC/n=5 | 70.4% | 66.7% | 80.5% | 52.1% | 81.5% |
| GWO | **72.7%** | **67.4%** | **84.5%** | **52.9%** | **88.0%** |

Table 3: Accuracy on GSM8K comparing GPT-4 and GPT-4o-mini.

| Method | Model | GSM8K |
|---|---|---|
| CoMAT (Leang et al., 2024) | GPT-4 | 93.7 |
| CoT (Ranaldi et al., 2025) | GPT-4 | 94.5 |
| FCoT (Lyu et al., 2023) | GPT-4 | 95.0 |
| MathPrompter (Imani et al., 2023) | GPT-4 | 95.6 |
| QuaSAR (Radford et al., 2018) | GPT-4 | 96.5 |
| MathDivide (Srivastava & Gandhi, 2024) | GPT-4 | 96.8 |
| **GWO/n=6+CoT** | **GPT-4o-mini** | **96.5** |

In mathematical reasoning tasks, GWO outperforms baseline methods CoT and CoT-SC ($n = 5$) on multiple models. For instance, on GPT-4.1-mini, GWO achieves 96.9% on GSM8K (compared to CoT's 88.2% and CoT-SC's 91.8%) and 92.7% on SVAMP (compared to CoT's 86.1% and CoT-SC's 87.9%); on Qwen2.5-Coder-7B-Instruct, GWO reaches 89.1%, 72.1%, and 62.3% on GSM8K, MATH, and AQUA, respectively, outperforming CoT (77.3%, 69.7%, 60.4%) and CoT-SC (80.1%, 71.2%, 62.1%); on Gemma-3-12b-it, GWO reaches 92.8% on GSM8K and 80.1% on MATH, also outperforming the baseline methods.

Table 4: Accuracy (%) on GSM8K and MATH using Qwen2.5-Coder-7B.

| Method | Base | GSM8K | MATH |
|---|---|---|---|
| OMI2 (Li et al., 2025) | Qwen2.5 | 84.1 | 72.3 |
| CODEI/O++ (Li et al., 2025) | Qwen2.5 | 85.7 | 72.1 |
| PyEdu (Li et al., 2025) | Qwen2.5 | 85.8 | 71.4 |
| CODEI/O (Li et al., 2025) | Qwen2.5 | 86.4 | 71.9 |
| OC-SFT-1 (Li et al., 2025) | Qwen2.5 | 86.7 | 70.9 |
| WI (Li et al., 2025) | Qwen2.5 | 87.0 | 71.4 |
| WI (Full) (Li et al., 2025) | Qwen2.5 | 87.0 | 71.1 |
| OMI2 (Full) (Li et al., 2025) | Qwen2.5 | 88.5 | 73.2 |
| CoT | Qwen2.5 | 77.3 | 69.7 |
| CoT-SC/n=5 | Qwen2.5 | 80.1 | 71.2 |
| GWO | Qwen2.5 | 89.1 | 72.1 |
| GWO+CoT | Qwen2.5 | 89.7 | 72.8 |
| **GWO/n=6+CoT** | **Qwen2.5** | **90.6** | **73.8** |

In knowledge and hybrid reasoning tasks, GWO demonstrates strong generalization ability. For instance, on GPT-4.1-mini, GWO achieves accuracies of 78.3%, 76.7%, and 78.7% on MMLU, BBH, and DATE, respectively—outperforming CoT (66.9%, 69.5%, 54.9%) and CoT-SC (71.2%, 72.6%, 57.9%). On Qwen2.5-Coder-7B-Instruct, GWO also surpasses baselines across MMLU (58.3%), BBH (53.9%), and CLUTRR (27.8%). Similarly, on Gemma-3-12b-it, it reaches 84.5% on DATE and 67.4% on BBH. To further assess GWO, we benchmark it on GSM8K and MATH using Qwen2.5-Coder-7B-Instruct. As shown in Table 4, GWO achieves 90.6% and 73.8%, outperforming mainstream optimization methods. Finally, integrating GWO ($n = 6$) with CoT yields strong results even when compared to GPT-4. As shown in Table 3, the combined model achieves 96.5% on GSM8K and 81.5% on MATH, matching or exceeding GPT-4's performance under various optimized settings.

GWO consistently improves over CoT and CoT-SC across models and tasks, showing both stronger reasoning accuracy and better generalization. Its complementarity with CoT further underscores its effectiveness as a lightweight yet powerful optimization strategy.

### 4.3 ADAPTABILITY EVALUATION

To further validate the generalization capability of the proposed GWO framework, we integrate it with mainstream Chain-of-Thought (CoT) reasoning and evaluate its performance across a wide range of reasoning tasks. CoT is first set as the baseline, and we then measure the improvements when GWO

is incorporated into CoT. Inference is conducted independently, and accuracy is computed using the official validation sets of each benchmark.

Table 5: Evaluation of adaptability of GWO to CoT on math reasoning task.

| Model | Math Reasoning Tasks | | | | | |
|---|---|---|---|---|---|---|
| | GSM8K | MATH | SVAMP | MultiArith | ASDiv | AQUA |
| **GPT-4o-mini** | | | | | | |
| CoT | 85.3% | 78.1% | 83.9% | 98.7% | 91.1% | 65.3% |
| GWO/n=5+CoT | 95.1% | 79.9% | 92.4% | 99.5% | 93.8% | 76.1% |
| GWO/n=6 | 95.7% | 80.6% | 92.6% | 99.7% | 94.1% | 76.4% |
| GWO/n=6+CoT | 96.5% | 81.5% | 92.9% | 99.8% | 94.5% | 76.8% |
| **GPT-4.1-mini** | | | | | | |
| CoT | 88.2% | 79.8% | 86.1% | 99.0% | 91.7% | 67.5% |
| GWO/n=5+CoT | 97.2% | 83.6% | 93.6% | 99.8% | 94.2% | 79.3% |
| GWO/n=6 | 97.9% | 83.9% | 94.2% | 99.9% | 94.5% | 79.7% |
| GWO/n=6+CoT | 98.3% | 84.3% | 94.8% | 99.9% | 94.8% | 80.4% |
| **GPT-4.1-nano** | | | | | | |
| CoT | 83.8% | 74.3% | 81.1% | 98.8% | 89.6% | 64.7% |
| GWO/n=5+CoT | 93.1% | 80.1% | 92.3% | 99.4% | 93.1% | 77.1% |
| GWO/n=6 | 93.8% | 80.6% | 92.9% | 99.4% | 93.4% | 77.6% |
| GWO/n=6+CoT | 94.2% | 81.2% | 93.8% | 99.7% | 94.0% | 78.2% |
| **Qwen2.5-Coder-7B-Instruct** | | | | | | |
| CoT | 77.3% | 69.7% | 82.1% | 92.1% | 86.4% | 60.4% |
| GWO/n=5+CoT | 89.7% | 72.8% | 90.6% | 97.7% | 90.5% | 62.5% |
| GWO/n=6 | 90.1% | 73.3% | 91.1% | 97.9% | 91.2% | 62.9% |
| GWO/n=6+CoT | 90.6% | 73.8% | 91.5% | 98.1% | 92.0% | 63.1% |
| **Gemma-3-12b-it** | | | | | | |
| CoT | 83.5% | 76.5% | 79.3% | 90.2% | 90.3% | 69.1% |
| GWO/n=5+CoT | 93.4% | 80.7% | 91.3% | 96.2% | 92.1% | 79.3% |
| GWO/n=6 | 93.7% | 81.2% | 91.8% | 96.5% | 92.6% | 79.9% |
| GWO/n=6+CoT | 94.3% | 81.3% | 92.4% | 96.8% | 93.1% | 80.5% |

Table 6: Evaluation of adaptability of GWO to CoT on hybrid reasoning tasks.

| Model | General | Multitask | Temporal | Logical | Math Mix |
|---|---|---|---|---|---|
| | MMLU | BBH | Date | CLUTRR | MATH_MIX |
| **GPT-4o-mini** | | | | | |
| CoT | 62.8% | 66.3% | 51.7% | 66.1% | 82.5% |
| GWO/n=5+CoT | 73.7% | 71.3% | 77.1% | 74.9% | 85.0% |
| GWO/n=6 | 74.1% | 71.5% | 77.8% | 75.4% | 87.5% |
| GWO/n=6+CoT | 74.9% | 72.2% | 78.3% | 75.8% | 95.1% |
| **GPT-4.1-mini** | | | | | |
| CoT | 66.9% | 69.5% | 54.9% | 71.2% | 82.5% |
| GWO/n=5+CoT | 78.9% | 77.2% | 79.3% | 78.8% | 85.0% |
| GWO/n=6 | 79.5% | 77.8% | 79.9% | 79.1% | 92.0% |
| GWO/n=6+CoT | 79.8% | 78.3% | 80.5% | 79.5% | 97.4% |
| **GPT-4.1-nano** | | | | | |
| CoT | 61.3% | 65.7% | 52.1% | 64.2% | 80.0% |
| GWO/n=5+CoT | 74.1% | 70.3% | 72.3% | 72.3% | 83.0% |
| GWO/n=6 | 74.3% | 70.8% | 78.2% | 72.9% | 88.0% |
| GWO/n=6+CoT | 75.1% | 71.2% | 79.3% | 73.5% | 93.5% |
| **Qwen2.5-Coder-7B-Instruct** | | | | | |
| CoT | 55.1% | 47.2% | 31.1% | 20.2% | 73.5% |
| GWO/n=5+CoT | 58.5% | 54.8% | 37.7% | 28.1% | 75.5% |
| GWO/n=6 | 58.9% | 55.3% | 38.3% | 28.8% | 83.5% |
| GWO/n=6+CoT | 59.1% | 55.6% | 39.2% | 26.1% | 90.2% |
| **Gemma-3-12b-it** | | | | | |
| CoT | 68.3% | 64.1% | 77.9% | 49.3% | 78.0% |
| GWO/n=5+CoT | 73.1% | 68.1% | 85.3% | 53.5% | 81.5% |
| GWO/n=6 | 73.6% | 68.3% | 85.8% | 54.1% | 88.0% |
| GWO/n=6+CoT | 74.2% | 68.8% | 86.7% | 54.4% | 94.3% |

**Mathematical reasoning tasks.** As shown in Table 5, GWO consistently boosts the accuracy of CoT across diverse mathematical reasoning datasets, including GSM8K, MATH, SVAMP, MultiArith, ASDiv, and AQUA. For example, on GPT-4o-mini, baseline CoT achieves accuracies of 85.3%, 78.1%, and 65.3% on GSM8K, MATH, and AQUA, respectively. With GWO($n = 5$)+CoT, the scores increase to 95.1%, 79.9%, and 76.1%, and further improve to 96.5%, 81.5%, and 76.8% when $n = 6$. Similar trends are observed on GPT-4.1-mini and GPT-4.1-nano, where the improvements are even more pronounced. For instance, on GPT-4.1-mini, GSM8K accuracy rises from 88.2% (CoT) to 98.3% (GWO($n = 6$)+CoT). Smaller models such as Qwen2.5-Coder-7B-Instruct and Gemma-3-12b-it also benefit considerably: on AQUA, Gemma's accuracy increases from 69.1% to 80.5%, while Qwen2.5's accuracy improves from 60.4% to 63.1% under the same setting. These results highlight the scalability of GWO, demonstrating effectiveness across both large and small-scale models.

**Hybrid reasoning tasks.** Table 6 summarizes the results on general-purpose and hybrid reasoning benchmarks, including MMLU, BBH, Temporal Reasoning (Date), CLUTRR, and MATH_MIX. Again, GWO provides consistent and significant gains over CoT. On GPT-4o-mini, baseline CoT achieves accuracies of 62.8%, 66.3%, and 51.7% on MMLU, BBH, and Temporal Reasoning, respectively. After integrating GWO($n = 6$), these scores increase to 74.9%, 71.5%, and 78.3%. Logical reasoning (CLUTRR) also benefits: GPT-4.1-mini improves from 71.2% (CoT) to 79.5% (GWO($n = 6$+CoT)). For composite mathematical reasoning (MATH_MIX), substantial improvements are observed, with GPT-4o-mini's accuracy increasing from 82.5% to 95.1% and GPT-4.1-mini's from 82.5% to 97.4%. Even on relatively weaker models such as Qwen2.5-Coder-7B-Instruct and Gemma-3-12b-it, GWO brings notable improvements, e.g., boosting Temporal Reasoning on Gemma from 77.9% (CoT) to 86.7% (GWO($n = 6$)).

Overall, the results clearly demonstrate that integrating GWO into CoT substantially enhances reasoning performance across diverse datasets and model architectures. The improvements span both mathematical and hybrid reasoning tasks, covering arithmetic, algebra, geometry, logic, temporal reasoning, and interdisciplinary benchmarks. Furthermore, the consistent gains across models of different sizes (from lightweight variants like GPT-4.1-nano to large-scale models like Gemma-3-12b-it) underscore the scalability, transferability, and robustness of the GWO framework in boosting reasoning accuracy and stability.

## 4.4 ABLATION STUDY

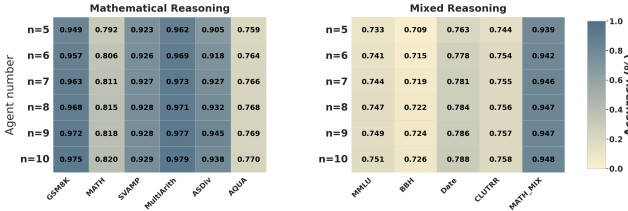

Figure 4: Accuracy over varying number of agents on mathematical (right) and hybrid (left) reasoning tasks. Increasing the number of agents generally improves accuracy.

To analyze the contribution of the number of agents and iterations in the GWO framework, we conduct an ablation study based on GPT-4o-mini across multiple datasets. We evaluate the impact of varying the number of agents, with $n \in \{5, 6, 7, 8, 9, 10\}$, on model performance. As shown in Figure 4, the accuracy across different mathematical reasoning datasets (GSM8K, MATH, SVAMP, MultiArith, ASDiv, and AQUA) generally improves with the increase of the number of agents. For instance, when $n = 5$, the accuracy on the GSM8K dataset is 94.9%, whereas it improves to 97.5% when $n = 10$. Similarly, in the mixed reasoning datasets such as MMLU, BBH, Date, CLUTRR, and MATH_MIX, the accuracy increases with the number of agents. For example, on the MMLU dataset, the accuracy increases from 73.3% at $n = 5$ to 75.1% at $n = 10$. Moderate increases yield limited, non-linear accuracy gains, requiring consideration of computational costs.

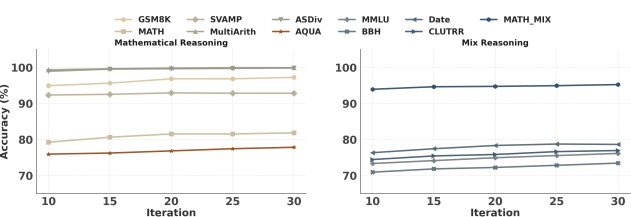

Figure 5: Accuracy over iteration for math (left) and hybrid (right) reasoning tasks when the number of agents is fixed to five. The accuracy increases with the number of iterations.

As shown in Figure 5, when the number of agents is fixed at $n = 5$, increasing the number of iterations from 10 to 30 consistently improves the accuracy of GWO across both mathematical and hybrid reasoning tasks. This indicates that additional iterations allow agents to explore the solution space more thoroughly and progressively refine their reasoning strategies. Figure 5 further confirms this trend at the dataset level, where nearly all benchmarks exhibit monotonic or near-monotonic improvements as the iteration count increases, highlighting the critical role of iterative optimization in enhancing overall performance.

Overall, both the number of agents and the number of iterations are key determinants of the effectiveness of GWO. A larger number of agents facilitates diverse exploration by introducing heterogeneous reasoning trajectories, while more iterations promote convergence stability and reduce variance across runs. Nevertheless, practical deployment requires balancing these benefits against computational cost and inference latency, in order to select an optimal configuration that achieves strong accuracy without excessive resource consumption. This trade-off is especially relevant when scaling GWO to larger models or real-world applications, where efficiency and responsiveness are as important as accuracy.

## 5 CONCLUSION

This paper introduces a novel multi-agent collaboration framework inspired by the Grey Wolf Optimizer (GWO), integrating Large Language Models (LLMs) with Multi-Agent Systems (MAS) to tackle complex coordination and optimization challenges. Leveraging GWO's hierarchical structure and dynamic "wolf pack" mechanism, the framework iteratively assigns roles and optimizes parameters to train agents, achieving superior task performance. Experimental results demonstrate the framework's strong adaptability across tasks of varying complexity. When combined with the Chain of Thought (CoT) method, it exhibits exceptional compositional capabilities. Furthermore, accuracy significantly improves with increased agent numbers and iterations. By enabling more efficient and interpretable multi-agent collaboration, the framework holds promise for advancing AI-driven solutions in society. This framework offers a promising solution for complex reasoning problems. Future work will focus on enhancing computational efficiency and exploring applications in dynamic knowledge management and decision-making.

## REPRODUCIBILITY STATEMENT

We are committed to the full reproducibility of this work. The proposed Grey Wolf Optimizer (GWO)-based multi-agent collaboration framework, along with its core algorithms and iterative optimization procedure, is fully described with pseudocode in the Appendix to ensure that researchers can directly reproduce and extend our study. Our experimental setup is detailed in Section 4. We evaluate the framework on both mathematical reasoning and hybrid reasoning benchmarks. These publicly available datasets cover arithmetic, algebra, geometry, logic, and interdisciplinary reasoning, thereby ensuring diverse experimental validation. For model selection, we employ several mainstream large language models to verify the robustness of our method across different scales and architectures. To guarantee transparency, the algorithm pseudocode (including the standard GWO and the agent-based iterative optimization process) is provided in Appendix A, while experimental results, ablation studies, and additional performance comparisons are presented in Appendix A.4 and A.5. All experiments are implemented in a Python environment and executed on a multi-GPU system to ensure efficiency in both inference and training. We will release the complete source code and configuration files upon publication, enabling other researchers to directly verify and extend our results.

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

# A    APPENDIX

## A.1    DETAILED RELATED WORK

### A.1.1    COLLABORATIVE REASONING WITH MULTI-AGENT LLMS

**Multi-Agent Systems (MAS)** originated from Distributed Artificial Intelligence (DAI), introducing core concepts such as autonomy, cooperation, and coordination. The Contract Net Protocol played a pivotal role in task allocation, establishing a theoretical foundation for MAS (Wooldridge & Jennings, 1995). Ferber et al. (Ferber & Weiss, 1999) further explored organizational paradigms, including holistic and emergent systems, demonstrating the scalability and robustness of MAS in domains such as robotics, logistics, and simulation. These systems have been increasingly applied to fields including game theory, distributed control, and collective collaboration (Wooldridge & Jennings, 1995; Ferber & Weiss, 1999; Yang & Wang, 2020; Stone & Veloso, 2000; Gerkey & Mataric, 2003; Guo et al., 2024; Li et al., 2024).

With the rapid advancement of Large Language Models (LLMs) (Naveed et al., 2023; Kumar, 2024), their powerful capabilities in language understanding and reasoning have injected new vitality into MAS. LLMs enable agents to execute complex tasks such as planning, programming, and verification through dynamic, natural language-driven interactions (Xi et al., 2025). For instance, the AutoGen framework (Wu et al., 2023) facilitates task decomposition among agents via prompt engineering and message passing, significantly improving collaboration efficiency. Furthermore, the Chain of Agents framework further addressed long-context tasks by organizing LLM agents into chains, thereby enhancing their ability to process complex information collaboratively (Zhang et al., 2024b).

Liu et al. (Liu et al., 2025) proposed an LLM-based agent framework incorporating brain, perception, and action modules, highlighting the effectiveness of specialized agents and interactive mechanisms in solving complex problems. Du et al. (Du et al., 2023) demonstrated the potential of LLMs in collaborative reasoning by improving MMLU task performance through multi-round debates.

Regarding coordination and communication, Distributed Constraint Optimization Problems (DCOP) address multi-agent constraint solving through negotiation and are widely used in task allocation (Fioretto et al., 2018). The FIPA-ACL protocol provides a standardized framework for agent communication (Fipa, 2002), while attention-based communication models have shown significant advantages in improving collaborative efficiency (Jiang & Lu, 2018). Recent research on MacNet (Qian

et al., 2024), a network built upon a directed acyclic graph topology, reveals that irregular structures outperform regular ones and proposes a logistic growth law for collaboration performance as the number of agents increases.

Furthermore, LLM-MAS has demonstrated broad application prospects in various domains. In the sandbox environment of Smallville, agents exhibit human-like collaborative behavior by planning schedules, sharing information, and coordinating activities through social interactions (Park et al., 2023). In the field of recommender systems, the Agent4Rec platform simulates user interactions and reveals phenomena such as the "filter bubble" (Zhang et al., 2024a). In intelligent manufacturing, an LLM-enhanced system framework leverages natural language communication to autonomously assign G-code tasks, significantly improving flexibility in production processes (Lim et al., 2024). In macroeconomic simulations, LLM agents exhibit heterogeneity in work and consumption behaviors and successfully replicate classical economic patterns such as the Phillips curve (Li et al., 2023).

### A.1.2    Swarm Intelligence Optimization Algorithms

**Swarm Intelligence Algorithms (SIA)** Chakraborty & Kar (2017); Slowik & Kwasnicka (2017); Tang et al. (2021); Priyadarshi & Kumar (2025) solve complex optimization problems by simulating the collaborative behavior of biological swarms in nature, such as ant colonies and bird flocks. These algorithms emphasize how simple agents can achieve complex global behavior through local interactions. The Particle Swarm Optimization (PSO) proposed by Kennedy et al. simulates the foraging behavior of bird flocks through velocity-position updates, making it suitable for continuous optimization Kennedy & Eberhart (1995), while Dorigo's Ant Colony Optimization (ACO), inspired by pheromone communication, excels in discrete optimization problems such as the Traveling Salesman Problem Dorigo (1992). These foundational works laid the theoretical groundwork for SIA.

In recent years, the integration of SIA with Deep Learning (DL) and Large Language Models (LLMs) has emerged as a research hotspot. Wang et al. proposed Deep PSO (DPSO), which embeds deep neural networks into the PSO framework to optimize neural network parameters, thereby improving high-dimensional feature selection performance Wang et al. (2018). Shriyan et al. combined PSO with LLMs to optimize prompt engineering, significantly enhancing the performance of LLMs on complex tasks Shriyan et al.. Furthermore, the hybrid GA-PSO algorithm by Sheikhalishahi et al. enhances population diversity through genetic operations, alleviating the issue of premature convergence in high-dimensional optimization problems Sheikhalishahi et al. (2013).

SIA has also been widely applied in engineering optimization, energy management, and data mining. For instance, Li et al. improved ACO to address multi-objective vehicle routing problems Li et al. (2019); Paliwal et al. employed the Artificial Bee Colony (ABC) algorithm to optimize microgrid scheduling Paliwal et al. (2020); Tu et al. combined PSO with Support Vector Machines (SVM) to enhance feature selection accuracy Tu et al. (2007). Despite their strong performance in various fields, SIA still faces challenges such as local optima in high-dimensional spaces, dependence on empirically tuned hyperparameters, and insufficient real-time adaptability in dynamic optimization problems Mavrovouniotis et al. (2017); Tang et al. (2021). Moreover, support for multi-objective optimization remains limited Janga Reddy & Nagesh Kumar (2007).

**The Grey Wolf Optimizer (GWO)** Mirjalili et al. (2014), proposed by Mirjalili et al., simulates the hierarchical structure (alpha, beta, delta, and omega wolves) and hunting behavior of grey wolves. It achieves global optimization by combining leadership guidance with random exploration. Compared to PSO and ACO, GWO features fewer hyperparameters and faster convergence, making it well-suited for continuous optimization. However, it is prone to local optima in high-dimensional problems.

To overcome these limitations, various improvements have been proposed. For example, Meidani et al. introduced adaptive GWO by incorporating dynamic parameter adjustment to enhance convergence performance Meidani et al. (2022); Wang et al. designed a hybrid GWO (HGWO) that integrates Differential Evolution (DE) with GWO to improve population diversity and global search capability Wang et al. (2022b); Zhang et al. proposed chaotic GWO (CGWO), which uses chaotic mapping to optimize initialization and step size, improving performance on high-dimensional optimization problems Zhang et al. (2021).

GWO has been widely applied in multiple domains. In engineering optimization, GWO has been used to solve economic dispatch problems in power systems by optimizing generation cost and

emissions to achieve multi-objective optimization Pradhan et al. (2016). In machine learning, GWO has been combined with SVM for feature selection to enhance classification accuracy in breast cancer diagnosis Kamel et al. (2019). In natural language processing, GWO has been employed to optimize the hyperparameters of LSTM language models, significantly improving their modeling performance Aufa et al. (2020). Moreover, GWO has been applied to image processing, where it is used to optimize image segmentation thresholds, thereby improving the segmentation quality of medical images Hu et al. (2024). Additionally, GWO has been applied to vehicle routing problems (VRP), optimizing route planning to reduce logistics costs Hussein et al. (2023).

## A.2 Notation

| Symbol | Definition |
|---|---|
| **Grey Wolf Optimizer (GWO) Symbols** | |
| $N$ | Population size (number of wolves/solutions) |
| $D$ | Search space dimensionality |
| $\mathbf{X}_i$ | Position of the $i$-th wolf: $\mathbf{X}_i = (x_i^1, x_i^2, \ldots, x_i^D)$ |
| $f(\mathbf{X})$ | Objective function to be optimized |
| $T_{\max}$ | Maximum number of iterations |
| $t$ | Current iteration number |
| $\mathbf{X}_p(t)$ | Prey position at iteration $t$ (current best solution) |
| $\mathbf{C}$ | Coefficient vector, $\mathbf{C} = 2\mathbf{r}_1$, $\mathbf{r}_1 \sim \mathcal{U}(0,1)^D$ |
| $\mathbf{D}$ | Distance vector, $\mathbf{D} = |\mathbf{C} \cdot \mathbf{X}_p(t) - \mathbf{X}_i(t)|$ |
| $\mathbf{A}$ | Adaptive coefficient, $\mathbf{A} = 2\mathbf{a} \cdot \mathbf{r}_2 - \mathbf{a}$, $\mathbf{r}_2 \sim \mathcal{U}(0,1)^D$ |
| $\mathbf{a}$ | Linearly decreasing parameter, $\mathbf{a} = 2 - 2 \cdot \frac{t}{T_{\max}}$ |
| $\alpha, \beta, \delta$ | The top three wolves (leaders) in GWO |
| $\omega$ | The remaining wolves (followers) |
| $\mathbf{D}_j$ | Distance from wolf $i$ to leader $j$: $\mathbf{D}_j = |\mathbf{C}_j \cdot \mathbf{X}_j - \mathbf{X}_i|, j \in \{\alpha, \beta, \delta\}$ |
| $\mathbf{A}_j, \mathbf{C}_j$ | Adaptive/random coefficients for leader $j$ |
| $\mathbf{X}_j'$ | Updated position guided by leader $j$ |
| $\mathcal{F}$ | Fitness function |
| **Agent Symbols** | |
| $n$ | Number of agents |
| $\text{Agent}_j$ | The $j$-th agent, $\text{Agent}_j = \{\text{LLM}_j, \text{clue}_j\}$ |
| $\text{LLM}_j$ | Large language model of agent $j$, $\text{LLM}_j = \{\boldsymbol{\eta}_j, \boldsymbol{\theta}\}$ |
| $\boldsymbol{\theta}$ | Shared model parameter set of LLMs |
| $\boldsymbol{\eta}_j$ | Agent-specific hyperparameter set: $\{T_j, p_j, F_j, E_j, M_j\}$ |
| $T_j$ | Temperature hyperparameter for agent $j$ |
| $p_j$ | Top-$p$ threshold for agent $j$ |
| $F_j$ | Frequency penalty for agent $j$ |
| $E_j$ | Presence penalty for agent $j$ |
| $M_j$ | Maximum token length for agent $j$ |
| $\text{clip}(x, [a,b])$ | Clipping function: $\max(a, \min(x, b))$ |
| $\mathcal{N}(\mu, \sigma^2)$ | Normal distribution with mean $\mu$ and variance $\sigma^2$ |
| $\mathcal{U}(a, b)$ | Uniform distribution over $[a, b]$ |
| $\mathcal{M}$ | Discrete set of possible maximum token lengths |
| $c_M$ | Constant value for maximum token length (fixed-length tasks) |
| **Data and Task Symbols** | |
| $\mathcal{D}$ | Reasoning problem dataset, $\mathcal{D} = \{q_1, q_2, \ldots, q_N\}$ |
| $q$ | A single question from dataset $\mathcal{D}$ |
| $\text{CoT}_j$ | Chain of Thought generated by agent $j$ |
| $\text{Answer}_j$ | Final answer generated by agent $j$ |
| $f(\text{Agent}_j, q)$ | Output of agent $j$ on $q$: $(\text{CoT}_j, \text{Answer}_j)$ |
| **Optimization Process Symbols** | |
| $K$ | Number of optimization iterations in multi-agent GWO |
| $\mathcal{A}$ | Population set of agents |
| $\mathbf{w}$ | Weight vector for leaders: $\mathbf{w} = \{w_\alpha, w_\beta, w_\delta\}, w_\alpha > w_\beta > w_\delta, \sum w = 1$ |
| ClueAdaptation | Clue template adaptation function |
| $(\boldsymbol{\eta}^*, clue^*)$ | Optimal hyperparameters and clue template found |

## A.3 PSEUDOCODE

---

**Algorithm 1** Grey Wolf Optimizer (GWO) — concise

---

**Init:** Initialize population $\{\mathbf{X}_i\}_{i=1}^N \subset \mathbb{R}^D$ within bounds      // Candidate solutions
1  Set $a \leftarrow 2$      // Exploration weight
2  (Optional) $\mathbf{X}^\star \leftarrow \arg\min_{\mathbf{X}_i} \mathcal{F}(\mathbf{X}_i)$      // Best-so-far
3  **Loop for** $t = 1, \ldots, T_{\max}$**: for** $t = 1$ **to** $T_{\max}$ **do**
4  |  Evaluate $\mathcal{F}(\mathbf{X}_i)$ for all $i$      // Objective values
5  |  Identify $\alpha, \beta, \delta$ (top-3 by fitness)      // Leaders
6  |  (Optional) update $\mathbf{X}^\star$  $a \leftarrow 2 - 2t/T_{\max}$      // Linear decrease
7  |  **foreach** $\mathbf{X}_i$ **do**
8  |  |  **foreach** $\ell \in \{\alpha, \beta, \delta\}$ **do**
9  |  |  |  Sample $\mathbf{r}_1, \mathbf{r}_2 \sim \mathcal{U}(0,1)^D$      // i.i.d. per dim
10 |  |  |  $\mathbf{A}_\ell \leftarrow 2a\,\mathbf{r}_1 - a$      // Coefficient
11 |  |  |  $\mathbf{C}_\ell \leftarrow 2\,\mathbf{r}_2$      // Coefficient
12 |  |  |  $\mathbf{D}_\ell \leftarrow |\,\mathbf{C}_\ell \odot \mathbf{X}_\ell - \mathbf{X}_i\,|$      // Distance
13 |  |  |  $\mathbf{X}'_\ell \leftarrow \mathbf{X}_\ell - \mathbf{A}_\ell \odot \mathbf{D}_\ell$      // Guided move
14 |  |  $\mathbf{X}_i \leftarrow \frac{1}{3}\left(\mathbf{X}'_\alpha + \mathbf{X}'_\beta + \mathbf{X}'_\delta\right)$      // Averaging
15 |  |  Project/clip $\mathbf{X}_i$ to bounds      // Feasible
16 Evaluate all $\mathcal{F}(\mathbf{X}_i)$      // Final check
17 **return** best $\mathbf{X}_i$ (or $\mathbf{X}^\star$)      // Solution

---

**Algorithm 2** : Agent Iterative Optimization Algorithm Based on GWO

---

**Initialization:**
Set $\mathcal{A} \leftarrow \emptyset$      // Initialize empty agent population
18 **for** $j = 1$ **to** $n$ **do**
19 |  $\boldsymbol{\eta}_j \leftarrow \texttt{SampleNormal}(\boldsymbol{\mu}, \boldsymbol{\sigma}^2)$      // Sample hyperparameters from normal distribution
20 |  $\boldsymbol{\eta}_j \leftarrow \texttt{Clip}(\boldsymbol{\eta}_j)$      // Ensure valid hyperparameter values
21 |  $clue_j \leftarrow \texttt{SampleClueTemplate}()$      // Generate clue template
22 |  $Agent_j \leftarrow (\boldsymbol{\eta}_j, clue_j)$      // Create agent
23 |  $\mathcal{A} \leftarrow \mathcal{A} \cup \{Agent_j\}$      // Add to population
24 **Optimization Loop:**
25 **for** \_ $= 1$ **to** $K$ **do**
26 |  **foreach** $Agent_j \in \mathcal{A}$ **do**
27 |  |  $q \leftarrow \texttt{SampleQuestion}(\mathcal{D})$
28 |  |  $(CoT_j, Ans_j) \leftarrow \texttt{LLM}(Agent_j, q)$
29 |  |  $Fitness_j \leftarrow \texttt{Evaluate}(Ans_j, q)$      // Stores agent scores
30 |  $(\alpha, \beta, \delta) \leftarrow \texttt{SelectTopAgents}(\mathcal{A}, Fitness)$      // Select top-3 agents as leaders
31 |  $\mathbf{w} \leftarrow \{w_\alpha, w_\beta, w_\delta\}$      // $w_\alpha > w_\beta > w_\delta$, $\sum w = 1$
32 |  **foreach** $Agent_j \in \mathcal{A} \setminus \{\alpha, \beta, \delta\}$ **do**
33 |  |  $r \sim \{\alpha, \beta, \delta\}$ with $\mathbf{w}$      // Sample by $\mathbf{w}$
34 |  |  $\boldsymbol{\eta}_j \leftarrow \texttt{Clip}(\texttt{SampleNormal}(\boldsymbol{\eta}_r, \sigma^2))$      // Mutate hyperparameters from leader
35 |  |  $clue_j \leftarrow \texttt{ClueAdaptation}(clue_j, \{clue_\alpha, clue_\beta, clue_\delta\})$      // Adapt clue using leader templates
36 $\boldsymbol{\eta}^* \leftarrow \boldsymbol{\eta}_\alpha$, $clue^* \leftarrow clue_\alpha$      // Extract best configuration
37 **return** $(\boldsymbol{\eta}^*, clue^*)$

---

## A.4 MORE EXPERIMENTAL RESULTS

The table 7compares diverse methods across multiple base models on the GSM8K and MATH benchmarks. Many advanced techniques, especially those leveraging large-scale models like Llama and Qwen, demonstrate competitive results. In particular, our proposed GWO / n = 6 + CoT method, when combined with GPT-4o-mini, achieves state-of-the-art performance, outperforming most existing approaches. This indicates the potential of our method to improve mathematical problem-solving capabilities while also acknowledging the strong foundation established by prior works.

Table 7: This table presents the accuracy of various methods on the GSM8K and MATH datasets. It compares different techniques applied to a range of base models, including Mistral, Code-Llama, Llama, Qwen, and GPT-4.

| Method | Base Model | GSM8k | MATH |
|---|---|---|---|
| No-CoT (Deng et al., 2024) | Mistral-7B | 38.0% | - |
| ICoT-SI (Deng et al., 2024) | Mistral-7B | 51.0% | - |
| - | RecurrentBlock-3.5B | 42.1% | - |
| MathCoder-CL (Wang et al., 2023a) | Code-Llama-7B | 67.8% | 30.2% |
| MAmmoTH (Yue et al., 2023) | Code-Llama-7B | 59.4% | - |
| Brain (Chen et al., 2024a) | Code-Llama-7B | 74.0% | - |
| SQ-VAE (Wang et al., 2023b) | Llama-2-7B | 40.0% | 7.0% |
| Self-Rewarding (Chen et al., 2024b) | Llama-2-7B | 40.0% | 10.7% |
| STaR (Zelikman et al., 2022) | Llama-2-7B | 58.2% | 16.0% |
| ENVISIONS (Xu et al., 2024) | Llama-2-7B | 59.0% | 19.0% |
| MetaMath (Yu et al., 2023b) | Llama-2-7B | 66.5% | - |
| ToRA-Code (Gou et al., 2023) | Llama-2-7B | 72.6% | - |
| OVM (Yu et al., 2023a) | Llama-2-7B | 73.7% | - |
| - | Llama-3.1-8B | 56.7% | 20.3% |
| - | Llama-3.1-70B | 85.5% | 41.4% |
| - | Llama-3.1-405B | 89.0% | 53.8% |
| - | NuminaMath-7B-CoT | 75.4% | 55.2% |
| - | DeepSeek-Coder-7B | 77.4% | 44.4% |
| - | Qwen2-7B | 79.9% | 44.2% |
| - | Qwen2-Math-7B | 80.4% | 50.4% |
| SIaM (Yu et al., 2024) | Qwen-2-Math-Base | 81.5% | 50.0% |
| - | Internlm2-math-plus-7B | 84.0% | 54.4% |
| OMI2 (Li et al., 2025) | Qwen2.5-Coder-7B | 84.1% | 72.3% |
| CODEI/O++ (Li et al., 2025) | Qwen2.5-Coder-7B | 85.7% | 72.1% |
| PyEdu (Li et al., 2025) | Qwen2.5-Coder-7B | 85.8% | 71.4% |
| CODEI/O (Li et al., 2025) | Qwen2.5-Coder-7B | 86.4% | 71.9% |
| OC-SFT-1 (Li et al., 2025) | Qwen2.5-Coder-7B | 86.7% | 70.9% |
| WI (Li et al., 2025) | Qwen2.5-Coder-7B | 87.0% | 71.4% |
| WI (Full) (Li et al., 2025) | Qwen2.5-Coder-7B | 87.0% | 71.1% |
| OMI2 (Full) (Li et al., 2025) | Qwen2.5-Coder-7B | 88.5% | 73.2% |
| CoT | Qwen2.5-Coder-7B | 77.3% | 69.7% |
| CoT | Qwen2.5-Coder-7B | 77.3% | 69.7% |
| CoT-SC/n=5 | Qwen2.5-Coder-7B | 80.1% | 71.2% |
| GWO | Qwen2.5-Coder-7B | 89.1% | 72.1% |
| GWO+CoT | Qwen2.5-Coder-7B | 89.7% | 72.8% |
| CoMAT (Leang et al., 2024) | GPT-4 | 93.7% | - |
| CoT (Ranaldi et al., 2025) | GPT-4 | 94.5% | - |
| FCoT (Lyu et al., 2023) | GPT-4 | 95.0% | - |
| MathPrompter (Imani et al., 2023) | GPT-4 | 95.6% | - |
| QuaSAR (Radford et al., 2018) | GPT-4 | 96.5% | - |
| MathDivide (Srivastava & Gandhi, 2024) | GPT-4 | 96.8% | - |
| **GWO/n=6+CoT** | **GPT-4o-mini** | **96.5%** | **81.5%** |

Table 8: Math Reasoning Performance

| Method | Math Reasoning | | | | | |
|---|---|---|---|---|---|---|
| | GSM8K | MATH | SVAMP | MultiArith | ASDiv | AQUA |
| **GPT-4o-mini** | | | | | | |
| CoT | 85.3% | 78.1% | 83.9% | 98.7% | 91.1% | 65.3% |
| CoT-SC/n=5 | 90.2% | 81.7% | 85.8% | 99.1% | 92.8% | 70.1% |
| GWO | 94.9% | 79.2% | 92.3% | 99.3% | 94.4% | 75.9% |
| GWO+CoT | 95.1% | 79.9% | 92.4% | 99.5% | 94.6% | 76.1% |
| GWO/n=6 | 95.7% | 80.6% | 92.6% | 99.7% | 94.8% | 76.6% |
| GWO/n=6+CoT | 96.5% | 81.5% | 92.9% | 99.8% | 94.9% | 76.8% |
| **GPT-4.1-mini** | | | | | | |
| CoT | 88.2% | 79.8% | 86.1% | 99.0% | 91.7% | 67.5% |
| CoT-SC/n=5 | 91.8% | 83.2% | 87.9% | 99.4% | 93.2% | 71.9% |
| GWO | 96.9% | 83.1% | 92.7% | 99.6% | 94.6% | 78.5% |
| GWO+CoT | 97.2% | 83.6% | 93.6% | 99.8% | 94.8% | 79.3% |
| GWO/n=6 | 97.9% | 83.9% | 94.2% | 99.9% | 94.9% | 79.7% |
| GWO/n=6+CoT | 98.3% | 84.3% | 94.8% | 99.9% | 94.9% | 80.4% |
| **GPT-4.1-nano** | | | | | | |
| CoT | 83.8% | 74.3% | 81.1% | 98.8% | 89.6% | 64.7% |
| CoT-SC/n=5 | 87.7% | 77.4% | 83.1% | 99.2% | 91.2% | 68.5% |
| GWO | 92.5% | 79.4% | 91.8% | 99.3% | 93.1% | 76.4% |
| GWO+CoT | 93.1% | 80.1% | 92.3% | 99.4% | 93.4% | 77.1% |
| GWO/n=6 | 93.8% | 80.6% | 92.9% | 99.4% | 93.7% | 77.6% |
| GWO/n=6+CoT | 94.2% | 81.2% | 93.8% | 99.7% | 94.0% | 78.2% |
| **Qwen2.5-Coder-7B-Instruct** | | | | | | |
| CoT | 77.3% | 69.7% | 82.1% | 92.2% | 86.4% | 60.4% |
| CoT-SC/n=5 | 80.1% | 71.2% | 84.2% | 95.1% | 88.7% | 62.1% |
| GWO | 89.1% | 72.1% | 90.1% | 97.1% | 91.2% | 62.3% |
| GWO+CoT | 89.7% | 72.8% | 90.6% | 97.7% | 92.0% | 62.5% |
| GWO/n=6 | 90.1% | 73.3% | 91.1% | 97.9% | 92.6% | 62.9% |
| GWO/n=6+CoT | 90.6% | 73.8% | 91.5% | 98.1% | 93.1% | 63.1% |
| **Gemma-3-12b-it** | | | | | | |
| CoT | 83.5% | 76.5% | 79.3% | 90.2% | 90.3% | 69.1% |
| CoT-SC/n=5 | 85.8% | 78.7% | 81.1% | 93.3% | 92.1% | 71.4% |
| GWO | 92.8% | 80.1% | 90.9% | 95.9% | 93.7% | 78.5% |
| GWO+CoT | 93.4% | 80.7% | 91.3% | 96.2% | 94.0% | 79.3% |
| GWO/n=6 | 93.7% | 81.2% | 91.8% | 96.5% | 94.3% | 79.9% |
| GWO/n=6+CoT | 94.3% | 81.3% | 92.4% | 96.8% | 94.6% | 80.5% |

Table 9: Other Datasets Performance

| Method | Gene. | Multitask | Temporal | Log. | Math Mix |
|---|---|---|---|---|---|
| | MMLU | BBH | Date | CLUTRR | MATH_MIX |
| **GPT-4o-mini** | | | | | |
| CoT | 62.8% | 66.3% | 51.7% | 66.1% | - |
| CoT-SC/n=5 | 67.6% | 68.9% | 54.7% | 72.6% | - |
| GWO | 73.3% | 70.9% | 76.3% | 74.4% | - |
| GWO+CoT | 73.7% | 71.3% | 77.1% | 74.9% | - |
| GWO/n=6 | 74.1% | 71.8% | 77.8% | 75.4% | - |
| GWO/n=6+CoT | 74.9% | 72.2% | 78.3% | 75.8% | 95.1% |
| **GPT-4.1-mini** | | | | | |
| CoT | 66.9% | 69.5% | 54.9% | 71.2% | - |
| CoT-SC/n=5 | 71.2% | 72.6% | 57.9% | 76.5% | - |
| GWO | 78.3% | 76.7% | 78.7% | 78.6% | - |
| GWO+CoT | 78.9% | 77.2% | 79.3% | 78.8% | - |
| GWO/n=6 | 79.5% | 77.8% | 79.9% | 79.1% | - |
| GWO/n=6+CoT | 79.8% | 78.3% | 80.5% | 79.5% | 97.4% |
| **GPT-4.1-nano** | | | | | |
| CoT | 61.3% | 65.7% | 52.1% | 64.2% | - |
| CoT-SC/n=5 | 65.8% | 68.2% | 55.6% | 69.3% | - |
| GWO | 73.5% | 69.1% | 77.1% | 71.6% | - |
| GWO+CoT | 74.1% | 70.3% | 77.8% | 72.3% | - |
| GWO/n=6 | 74.3% | 70.8% | 78.2% | 72.9% | - |
| GWO/n=6+CoT | 75.1% | 71.2% | 79.3% | 73.5% | 93.5% |
| **Qwen2.5-Coder-7B-Instruct** | | | | | |
| CoT | 55.1% | 47.2% | 31.1% | 20.2% | - |
| CoT-SC/n=5 | 56.1% | 49.5% | 32.9% | 21.1% | - |
| GWO | 58.3% | 53.9% | 37.1% | 27.8% | - |
| GWO+CoT | 58.5% | 54.8% | 37.7% | 28.1% | - |
| GWO/n=6 | 58.9% | 55.3% | 38.3% | 28.8% | - |
| GWO/n=6+CoT | 59.1% | 55.6% | 39.2% | 26.1% | 90.2% |
| **Gemma-3-12b-it** | | | | | |
| CoT | 68.3% | 64.1% | 77.9% | 49.3% | - |
| CoT-SC/n=5 | 70.4% | 66.7% | 80.5% | 52.1% | - |
| GWO | 72.7% | 67.4% | 84.5% | 52.9% | - |
| GWO+CoT | 73.1% | 68.1% | 85.3% | 53.5% | - |
| GWO/n=6 | 73.6% | 68.3% | 85.8% | 54.1% | - |
| GWO/n=6+CoT | 74.2% | 68.8% | 86.7% | 54.4% | 94.3% |

These tables 9 and tables 8 demonstrate that our proposed methods (GWO, GWO+CoT, GWO/n=6, GWO/n=6+CoT) achieve remarkable improvements across multiple math reasoning benchmarks and datasets. In GSM8K, GWO/n=6+CoT attains 96.5% (vs. 85.3% of CoT in GPT-4o-mini) and 98.3% (vs. 88.2% of CoT in GPT-4.1-mini), highlighting significant gains from structured reasoning. Similar trends are seen in SVAMP, AQUA, and Temporal Date. For instance, on GPT-4o-mini, GWO boosts Temporal Date performance from 51.7% (CoT) to 76.3%, with GWO/n=6+CoT increasing it further to 78.3%. The table shows these methods outperform baseline CoT and CoT-SC, enhancing math reasoning efficiently without high computational cost.

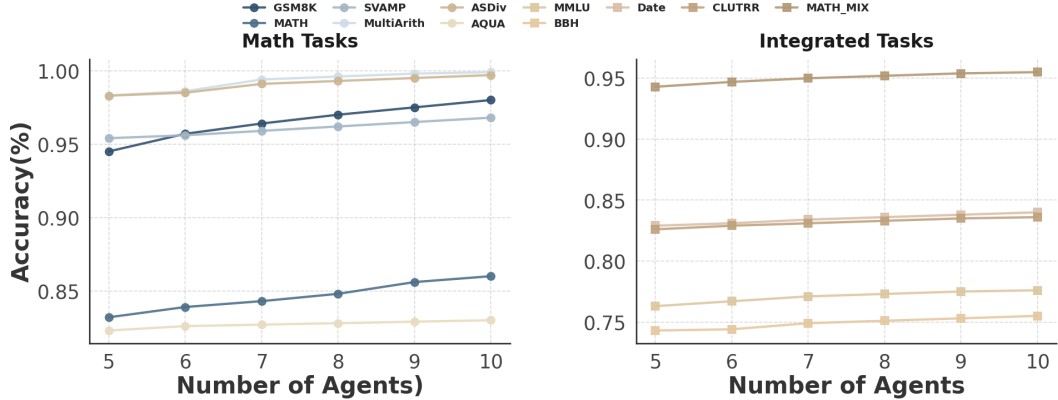

Figure 6: The figure displays two graphs comparing the accuracy of different tasks as the number of agents varies, with one graph focused on math-related tasks and the other on integrated tasks.

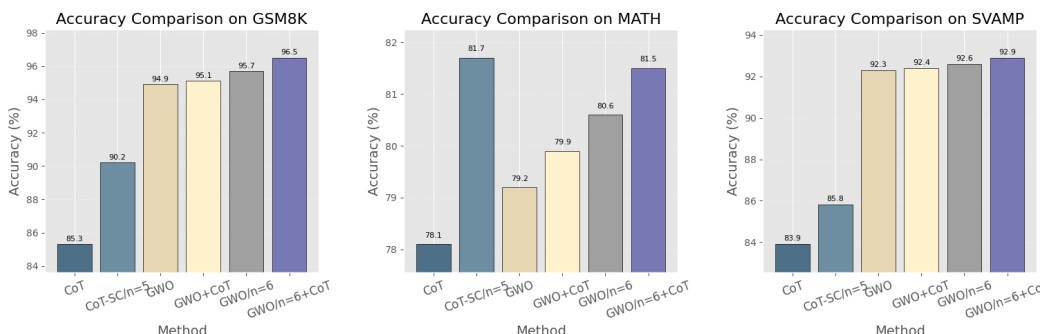

Figure 7: The figures present bar-charts comparing the accuracy of different methods across three datasets: GSM8K, MATH, and SVAMP. Each chart shows how various techniques perform in terms of accuracy, highlighting differences and trends among the methods for each respective dataset.

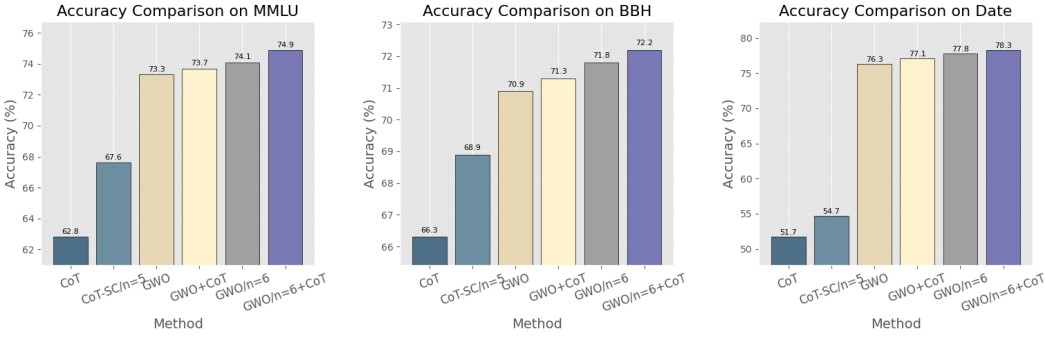

Figure 8: These figures are bar charts showing the accuracy comparison of different methods on datasets MMLU, BBH, and Date. Each chart visually represents how various techniques perform in terms of accuracy.

The bar chart illustrates the accuracy comparison on CLUTRR for different methods. Generally, there is an upward trend in accuracy. The "CoT" method shows the lowest accuracy at 66.1%. In contrast, methods like "CoT-SC/n = 5", "GWO", "GWO+CoT/n = 6", and "GWO/n = 6+CoT" demonstrate higher accuracies, with "GWO/n = 6+CoT" achieving the highest accuracy of 75.8%, indicating that incorporating certain techniques significantly improves performance.

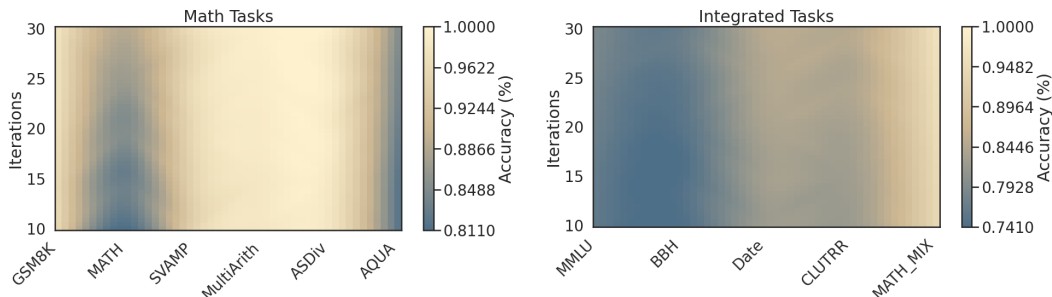

Figure 9: These two figures respectively show the accuracy of Math Tasks and Integrated Tasks at different iterations, with task names on the x-axis, iterations on the y-axis, and accuracy indicated by color gradients.

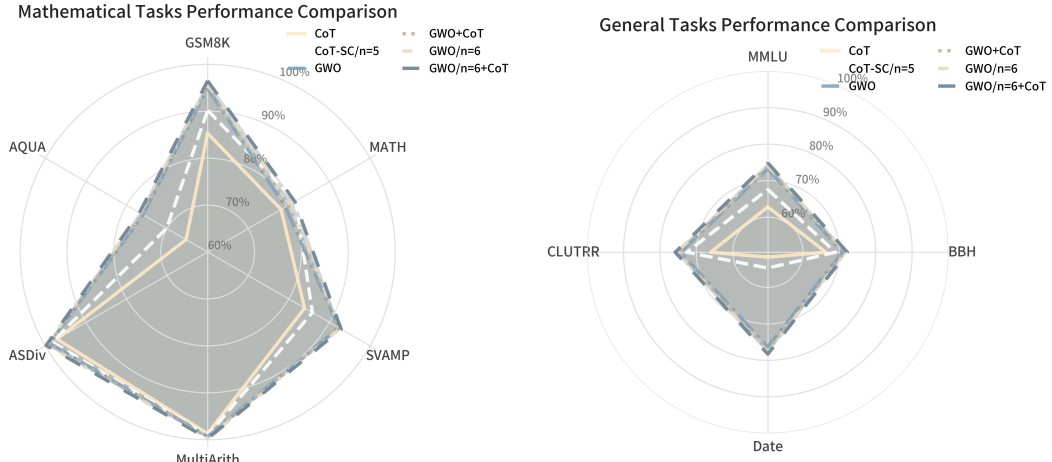

Figure 10: The first figure compares the performance of different methods on mathematical tasks, while the second figure contrasts their performance on general tasks, both using radar-chart visualizations.

## A.5 Training Examples

### A.5.1 Forward Inference Examples

---

**Dataloading for Interation 1**

**Problem:**          YYYY-MM-DD 22:35:34,676 - log.logger_utils - INFO

*Tiffany was collecting cans for recycling. On Monday, she had 3 bags of cans. The next day, she found 7 more bags of cans and 44 bags of bottles. How many bags of cans did she have altogether?*

**Answer:**          YYYY-MM-DD 22:35:34,676 - log.logger_utils - INFO

*10*

---

This example demonstrates the parameter settings, systematic problem-solving approach, and the beginning of the answer provided by the first agent when tackling a mathematical word problem. The focus is on the standardized process of problem-solving, emphasizing a complete workflow from understanding the question to checking the final answer.

---

**Training for Agent 1 in Interaction 1**

**Prameter Log for Agent 1:**    YYYY-MM-DD 22:35:34,691 - log.logger_utils - INFO

temperature: 0.9193923355128871,
top_p: 0.9027935433165666,
frequency_penalty: 1.8091973063219,
presence_penalty: -1.7443228132190376,
max_tokens: 1919

**Clue Log for Agent 1:**     YYYY-MM-DD 22:35:48,332 - log.logger_utils - INFO

To solve the questions found in the svamp dataset, follow these steps:
1. Carefully read the entire question to understand what is being asked.
2. Identify and extract all relevant numerical data and quantities mentioned within the question.
3. Determine which mathematical operations (addition, subtraction, multiplication, division) are necessary based on keywords or phrases that indicate relationships between numbers (e.g., "altogether," "more than," etc.).
4. Perform calculations step-by-step while keeping track of intermediate results if needed to avoid errors.
5. Double-check your final answer against the context of the problem to ensure it makes sense.
Use this approach systematically for each question from the SVAMP dataset you encounter!

**Answer by Agent 1:**      YYYY-MM-DD 22:35:52,418 - log.logger_utils - INFO

Let's follow the steps to solve the problem:
1. Read the entire question: Tiffany collected cans for recycling, and we need to find out how many bags of cans she had altogether after her collection over two days.
2. Extract relevant numerical data:
On Monday, Tiffany had 3 bags of cans.
On Tuesday, she found 7 more bags of cans.
3. Determine necessary mathematical operations: We need to find the total number of bags of cans, which involves addition since we are combining quantities.

---

4. Perform calculations step-by-step:
Start with the number of bags she had on Monday: 3
Add the number of additional bags found on Tuesday:

$$\text{Total Bags} = 3 + 7 = 10$$

5. Double-check final answer against context: The question asks for how many bags of "cans" she has altogether, and we calculated that Tiffany has a total of 10 bags (only counting "cans").
Therefore, Tiffany has a total of 10 bags of cans altogether.

This example provides a detailed demonstration of how the first agent systematically solves a mathematical word problem using a five-step approach (understanding the problem, extracting data, determining the operation, step-by-step calculation, and checking the answer), ultimately arriving at a clear and accurate solution.

**Training for Agent 2 in Interaction 1**

**Prameter Log for Agent 2:**     YYYY-MM-DD 22:35:52,532 - log.logger_utils - INFO
temperature: 0.39241099147650554,
top_p: 0.6602020470317261,
frequency_penalty: 0.00071422750930004872,
presence_penalty: -0.19136457572912707,
max_tokens: 1413

**Clue Log for Agent 2:**     YYYY-MM-DD 22:36:05,253 - log.logger_utils - INFO
Create a word problem that involves basic arithmetic operations based on the given scenario. The problem should include specific quantities and require calculations to determine totals, differences, or other values relevant to everyday activities, ensuring that it can engage learners in applying their math skills within relatable contexts like shopping, collecting items, or planning events.

**Answer by Agent 2:**     YYYY-MM-DD 22:36:09,262 - log.logger_utils - INFO
To find out how many bags of cans Tiffany had altogether, we need to focus only on the bags of cans.
1. On Monday, Tiffany had 3 bags of cans.
2. The next day, she found 7 more bags of cans.
Now, we can calculate the total number of bags of cans:
Total bags of cans = Bags on Monday + Bags found on Tuesday
Total bags of cans = 3 + 7
Now, let's do the calculation:
Total bags of cans = 3 + 7 = 10
So, Tiffany had 10 bags of cans altogether.
(Note: The bags of bottles are not included in the total for bags of cans, as the problem specifically asks for the total number of bags of cans.)

This example demonstrates how the second agent generates and solves a simple math word problem following a standardized process. It covers all key steps, including problem design, information extraction, mathematical operations, step-by-step calculation, and explanation of the result. The example also specifically clarifies the scope of the problem (counting only bags of cans, not bottles), which helps the agent understand the problem requirements and the solution approach.

**Training for Agent 3 in Interation 1**

**Prameter Log for Agent 3:**     YYYY-MM-DD 22:36:09,369 - log.logger_utils - INFO
temperature: 0.5822851007143794,

top_p: 0.9719972687061527,
frequency_penalty: 0.04468780736250461,
presence_penalty: 0.02409206816099846,
max_tokens: 1397

**Clue Log for Agent 3:**          YYYY-MM-DD 22:36:21,755 - log.logger_utils - INFO

You are given various scenarios involving numerical problems related to quantities, collections, and arithmetic operations. Your task is to answer questions based on the provided information for each scenario. Pay attention to details such as initial amounts, changes over time or circumstances (e.g., additions or subtractions), and specific requests regarding total counts. Use basic mathematical reasoning and calculations when determining your answers.

**Answer by Agent 3:**          YYYY-MM-DD 22:36:24,662 - log.logger_utils - INFO

To find out how many bags of cans Tiffany had altogether, we need to focus only on the bags of cans.
On Monday, Tiffany had 3 bags of cans. The next day, she found 7 more bags of cans.
Now, we can calculate the total number of bags of cans:
Total bags of cans = Bags on Monday + Bags found on Tuesday
Total bags of cans = 3 + 7 = 10
Therefore, Tiffany had a total of 10 bags of cans altogether.

Agent 3 focuses on generating simple and straightforward arithmetic word problems, usually centered on addition or subtraction, with scenarios drawn from everyday life for easy understanding. The problems and solutions are very concise, typically containing only key information without extra explanations or reasoning steps. This approach is suitable for foundational arithmetic instruction or quick practice, emphasizing the calculation itself rather than the problem-solving process.

**Training for Agent 4 in Interaction 1**

**Prameter Log for Agent 4:**          YYYY-MM-DD 22:36:24,768 - log.logger_utils - INFO

temperature: 0.5298405602481979,
top_p: 0.7372076916536928,
frequency_penalty: 0.02248299928407254,
presence_penalty: 0.049040988666425915,
max_tokens: 1746

**Clue Log for Agent 4:**          YYYY-MM-DD 22:36:37,855 - log.logger_utils - INFO

Create a word problem involving everyday scenarios that require basic arithmetic operations to solve. The scenario should include quantities related to common activities such as shopping, collecting items, or completing tasks over a period of time. Ensure the question asks for a total count based on additions and potential subtractions, utilizing numbers that are simple and relatable while encouraging mathematical reasoning in interpreting the context provided.

**Answer by Agent 4:**          YYYY-MM-DD 22:36:41,333 - log.logger_utils - INFO

To find out how many bags of cans she had altogether, we only need to focus on the bags of cans.
1. Start with the number of bags of cans Tiffany had on Monday: 3 bags.
2. Add the number of bags of cans she found on Tuesday: 7 bags.
Now, calculate the total number of bags of cans:

$$3 \text{ bags (Monday)} + 7 \text{ bags (Tuesday)} = 10 \text{ bags of cans}$$

Therefore, Tiffany had 10 bags of cans altogether.
(Note: The bags of bottles are not included in the total for cans.)

Agent 4's clues emphasize problem generation that incorporates real-life scenarios, encouraging mathematical reasoning and the filtering of relevant information. Its answers are usually broken down

into steps, highlighting the problem-solving approach and information extraction, but do not directly provide the final answer, focusing more on guiding the Agent to think independently. The overall style leans towards instructional guidance, helping the Agent understand the problem and develop independent problem-solving skills.

---

**Training for Agent 5 in Interation 1**

**Prameter Log for Agent 5:**     YYYY-MM-DD 22:36:41,435 - log.logger_utils - INFO

temperature: 0.3603354498081722,
top_p: 0.8427861799350946,
frequency_penalty: 0.0572741656361528,
presence_penalty: -0.124744427417488,
max_tokens: 1736

**Clue Log for Agent 5:**       YYYY-MM-DD 22:36:53,728 - log.logger_utils - INFO

Create a problem-solving question that involves basic arithmetic operations such as addition, subtraction, multiplication, or division within various contexts. The scenarios should include everyday situations involving quantities of objects or items to be calculated. Make sure the questions require an understanding of how to combine different amounts and solve for totals or differences based on the information provided in each scenario.

**Answer by Agent 5:**        YYYY-MM-DD 22:37:00,748 - log.logger_utils - INFO

To solve the problem, we need to focus only on the bags of cans, as the question specifically asks for the total number of bags of cans Tiffany had.
1. On Monday, Tiffany had 3 bags of cans.
2. On Tuesday, she found 7 more bags of cans.
Now, we can add the number of bags of cans she had on Monday to the number she found on Tuesday:

$$\text{Total bags of cans} = \text{Bags on Monday} + \text{Bags found on Tuesday}$$

Substituting the values:

$$\text{Total bags of cans} = 3 + 7$$

Now, we perform the addition:

$$\text{Total bags of cans} = 10$$

Thus, Tiffany had a total of 10 bags of cans altogether.

---

Agent 5's clues cover a variety of operations, including addition, subtraction, multiplication, and division, with a strong emphasis on information integration and analytical skills. Its solutions are very detailed, systematically presenting the problem-solving process step by step, including formulas, substitution, calculation, and conclusion, with formal and well-organized language.

### A.5.2 EVALUATING EXAMPLES

This section systematically evaluates the performance of the agent during a specific task iteration. The evaluation process adopts a structured scoring system, conducting detailed analysis and scoring across three core dimensions: logic, ingenuity, and completeness, ultimately resulting in a comprehensive total score.

---

**Evaluating for Agent 1 in Interation 1**

**Evalauation result:**        YYYY-MM-DD 22:37:10,792 - log.logger_utils - INFO

**Logic (0 - 100):**
"The solution follows a clear logical structure. It begins with understanding the question and identifying key data points, followed by performing the correct mathematical operations to

---

arrive at the answer. The steps are clearly articulated, ensuring that the reasoning is easy to follow."
(Score: 90)

**Ingenuity (0 - 100):**
"While the solution is logical and methodical, it lacks any innovative or creative approaches to solving the problem. It simply adheres to standard arithmetic operations without introducing any unique methods or insights."
(Score: 70)

**Completeness (0 - 100):**
"The answer comprehensively addresses all parts of the question, including reading the question, identifying numerical data, performing calculations, and verifying the final result. However, it could benefit from a brief summary reiterating the significance of reaching this total within the context."
(Score: 85)

**Total Score (0 - 100):** (Score: 82)

Specifically, the evaluator first reviews the agent's solution process to determine whether it follows clear logical steps, including understanding the problem, extracting key data points, and performing correct mathematical calculations; based on this, a score is given for the logic dimension. Next, the evaluator examines whether the agent demonstrates unique or innovative approaches in problem-solving, or simply relies on conventional methods, and assigns a score for ingenuity accordingly. Finally, the evaluator checks whether the answer fully addresses all aspects of the problem, including data identification, calculation, and result verification, as well as whether the significance of the result is summarized, thus determining the completeness score. Each dimension is accompanied by a brief written assessment highlighting strengths and weaknesses, and a comprehensive total score is calculated based on an overall consideration. This evaluation method provides a holistic reflection of the agent's problem-solving performance and offers concrete references for subsequent optimization and improvement.

**Evaluating for Agent 2 in Interaction 1**

**Evalauation result:**        YYYY-MM-DD 22:37:19,822 - log.logger_utils - INFO

**Logic (0 - 100):**
"The answer correctly identifies the number of bags of cans Tiffany had by logically breaking down the information provided. It accurately sums the bags from Monday and Tuesday without including irrelevant details about bottles."
(Score: 95)

**Ingenuity (0 - 100):**
"The approach is straightforward and effective, but it does not demonstrate any creative or innovative problem-solving techniques. The solution follows a standard arithmetic process without additional insights."
(Score: 70)

**Completeness (0 - 100):**
"The answer addresses the question fully by providing a clear calculation and stating that bags of bottles are not included in the total for bags of cans. However, it could have briefly summarized why only cans were considered to enhance clarity."
(Score: 85)

**Total Score (0 - 100):** (Score: 83)

**Evaluating for Agent 3 in Interation 1**

**Evalauation result:**  YYYY-MM-DD 22:37:27,859 - log.logger_utils - INFO

**Logic (0 - 100):**
"The solution correctly identifies the relevant information and follows a clear logical flow to arrive at the answer. The calculations are straightforward and accurate."
(Score: 90)

**Ingenuity (0 - 100):**
"The solution employs basic arithmetic to solve a simple problem. While it is effective, there is little creativity or innovation in the approach."
(Score: 60)

**Completeness (0 - 100):**
"The answer thoroughly addresses the problem, providing all necessary calculations and arriving at a final total. However, it could be enhanced by including a brief summary of the given information."
(Score: 85)

**Total Score (0 - 100):** (Score: 78)

**Evaluating for Agent 4 in Interation 1**

**Evalauation result:**  YYYY-MM-DD 22:37:37,127 - log.logger_utils - INFO

**Logic (0 - 100):**
"The reasoning presented in the answer is clear and follows a logical sequence. It identifies the relevant information (the number of bags of cans) and correctly excludes irrelevant information (the bags of bottles). The calculation is straightforward and accurate."
(Score: 90)

**Ingenuity (0 - 100):**
"The answer demonstrates a basic understanding of arithmetic and problem-solving. However, it lacks creative or innovative approaches to the problem, as it simply applies basic addition without any additional insights or unique methods."
(Score: 50)

**Completeness (0 - 100):**
"The answer provides a complete solution to the problem by addressing all necessary steps to arrive at the final count of bags of cans. It states clearly what is included in the count and what is not, making it easy to follow."
(Score: 90)

**Total Score (0 - 100):** (Score: 76)

**Evaluating for Agent 5 in Interation 1**

**Evalauation result:**  YYYY-MM-DD 22:37:37,229 - log.logger_utils - INFO

**Logic (0 - 100):**
"The solution follows a clear and logical sequence of steps to arrive at the answer. It correctly identifies the relevant information, uses appropriate arithmetic, and clearly states the final result."
(Score: 95)

**Ingenuity (0 - 100):**
"The approach is straightforward and relies on basic arithmetic, which does not showcase high levels of creativity or innovative problem-solving. However, it effectively solves the problem without unnecessary complexity."
(Score: 70)

**Completeness (0 - 100):**
"The answer provides a complete solution by including all necessary calculations and explanations. It explicitly states the initial quantities, performs the addition, and presents the final answer clearly."
(Score: 90)

**Total Score (0 - 100):** (Score: 85)

### A.5.3 OPTIMIZING EXAMPLES

This section illustrates the process of parameter updating during a particular iteration. Specifically, the system first ranks the agents participating in the current round of tasks and selects the top three performers. Next, the system updates the parameters of a certain agent (usually one that performed well or needs improvement). The parameter update involves adjusting key hyperparameters such as temperature, top_p, frequency_penalty, and presence_penalty, all of which directly affect the diversity, creativity, and accuracy of the agent's generated responses. The system records the specific values of these parameters before and after the update, and provides a task clue to clarify the type of problem the agent needs to solve and any important considerations.

---

**Ranking for Iteration 1**

**Ranking:**                              YYYY-MM-DD 22:37:54,082 - log.logger_utils - INFO

Top three results from message processing LLM:
"First Place": "2",
"Second Place": "5",
"Third Place": "1"

---

**Updating for Agent in Iteration 1**

**Before Updating:**                     YYYY-MM-DD 22:37:54,082 - log.logger_utils - INFO
Parameter:
temperature=0.9193923355128871,
top_p=0.9027935433165666,
frequency_penalty=1.8091973063219,
presence_penalty=-1.7443228132190376
Clue:
You are given various scenarios involving numerical problems related to quantities, collections, and arithmetic operations. Your task is to answer questions based on the provided information for each scenario. Pay attention to details such as initial amounts, changes over time or circumstances (e.g., additions or subtractions), and specific requests regarding total counts. Use basic mathematical reasoning and calculations when determining your answers.

**After Updating:**                      YYYY-MM-DD 22:38:02,220 - log.logger_utils - INFO
Parameter:
temperature=0.41004817918398145,
top_p=0.9247874518938444,
frequency_penalty=-0.0035787094690344773,
presence_penalty=0.025675460113628396
Clue:
You are presented with various scenarios that involve numerical problems related to quantities

> and arithmetic operations. Your task is to answer questions based on the provided information for each scenario. Start by carefully reading each question to understand what is being asked. Identify all relevant numerical data and quantities mentioned. Determine which mathematical operations (addition, subtraction, multiplication, division) are needed based on keywords or phrases that indicate relationships between numbers. Perform calculations step-by-step, keeping track of intermediate results if necessary, and double-check your final answer against the context of the problem to ensure it makes sense. Pay attention to details such as initial amounts, changes over time, and specific requests regarding total counts, using basic mathematical reasoning and calculations to determine your answers.

This entire process embodies an automated "evaluation–selection–fine-tuning–re-evaluation" optimization loop, aiming to continuously improve the agent's performance on specific types of problems through iterative trial and adjustment.

### A.6 ACKNOWLEDGE

This article used large language models (such as ChatGPT) as an auxiliary tool in the language polishing process, but did not use them in research conception and academic content generation.

