# OpenReview forum: "Agent-GWO: Collaborative Agents for Dynamic Prompt Optimization in Large Language Models"
_ICLR.cc/2026/Conference — ICLR 2026 Conference Withdrawn Submission_

### Official Review · Reviewer_yb2o · 2025-10-31

**Soundness:** 2
**Presentation:** 3
**Contribution:** 2
**Rating:** 4
**Confidence:** 3

**Summary:**

This paper proposes a novel dynamic prompt optimization framework termed ​​Agent-GWO​​, which enhances the performance of Large Language Models (LLMs) in complex reasoning tasks by integrating ​​multi-agent collaboration​​ with the ​​Grey Wolf Optimizer (GWO)​​. The framework dynamically refines task-specific prompts through iterative optimization, avoiding the high cost of model fine-tuning. Extensive experiments demonstrate its effectiveness across mathematical reasoning (e.g., GSM8K, MATH) and hybrid reasoning tasks (e.g., MMLU, BBH). However, several critical issues undermine the robustness and reproducibility of the claimed contributions, including insufficient hyperparameter specifications, limited baseline comparisons, lack of computational cost analysis, and theoretical gaps in convergence proofs.

**Strengths:**

1.First work to combine GWO's hierarchical optimization structure with multi-agent LLM systems, enabling dynamic prompt refinement through leader-follower mechanisms (α, β, δ wolves guiding ω agents).
2.Rigorous testing across 10+ datasets and 5 LLM architectures (including GPT-4o-mini and Qwen2.5-Coder), with ablation studies validating the impact of agent population size and iteration counts.
3.The parameter-agnostic approach avoids model fine-tuning, making it adaptable to diverse tasks ranging from mathematical reasoning to domain-specific applications.

**Weaknesses:**

1. Critical hyperparameters (e.g., μ, σ for normal distributions sampling temperature/top-p) lack specified values or justification, preventing experimental replication.
2. ONLY Comparisons limited to CoT and CoT-SC, omitting state-of-the-art dynamic methods (GoT, ToT) and reinforcement learning approaches, weakening validity claims. The baseline is weak now.
3.No analysis of token consumption or inference latency despite controlling agent/iteration counts, masking true deployment costs.

**Questions:**

1.What specific values were used for hyperparameter sampling (e.g., μ, σ in normal distributions), and how were these ranges empirically determined?
2.Why were contemporary dynamic optimization methods (GoT, ToT) excluded from comparisons, given their relevance to prompt engineering?
3.What is the token efficiency of Agent-GWO compared to baseline methods, and how does scalability impact real-world deployment?

---

### Official Review · Reviewer_3W3c · 2025-10-31

**Soundness:** 2
**Presentation:** 3
**Contribution:** 2
**Rating:** 4
**Confidence:** 4

**Summary:**

The paper presents a novel prompt optimization technique, relying on a swarm of dynamically updated problem-solving LLM-based agents, each of which is characterized by the prompt and the decoding parameters. On each iteration, top 3 agents are identified, and the rest are updated to closer match characteristics of the top ones – the optimization algorithm used is the Grey Wolf Optimizer (GWO) adapted from prior research.

**Strengths:**

1. In the context of prompt optimization, the idea of using a swarm of dynamically updated agents, each of which encapsulates not only prompt, but also decoding parameters, is both reasonable and novel.
2. The method is evaluated across a decent number of benchmarks and is shown to consistently outperform the simple CoT prompting
3. The method’s performance is shown to improve with the number of agents / iterations
4. The writing is, in general, good, and the illustrations are excellent

**Weaknesses:**

1. One major weakness is that the paper completely ignores currently existing prompt optimization techniques, which is a large body of work. While the method is definitely promising, it is hard to assess its actual usefulness without comparison with these techniques (see some citations below). Therefore, the third research contribution claimed (“Extensive experiments show that our method consistently surpasses strong baselines”) is not substantiated
2. The first contribution claim – “We find that dynamically optimizing prompts during training enables LLMs to develop more effective, task-specific strategies.” – does not seem to be fully correct either, as optimization is performed on the frozen LMs. However, jointly (post-)training the model and optimizing the prompt would be an interesting research direction indeed
3. The analysis of the prompt / decoding parameters evolution is completely missing from both the main paper and the appendix: there is not a single optimized prompt example present. One could even hypothesize that the performance improvement is due to the temperature associated with the agents converging to 0, rather than the prompts’ improvement
4. While jointly optimizing the prompt and the decoding parameters is reasonable, the potential of such “bundling” feels underexplored, i.e. different agents could be powered by different LMs
5. It is not clear why, instead of verifiable rewards (=accuracy), the evaluation part of the pipeline follows the LLM-as-a-judge approach. While this could be justified by richer textual rewards (“TextGrad: Automatic "Differentiation" via Text”, Yuksekgonul et al.; “TRPrompt: Bootstrapping Query-Aware Prompt Optimization from Textual Rewards”, Nica et al.), the textual feedback is rendered into the numeric score prior to the optimization step, so the benefits of textual feedback are never used, making the design choice of this pipeline component questionable. Furthermore, there is no analysis of evolution across different assessment dimensions. LLM-as-a-judge meta-prompt is also missing.
6. Some minor issues: (i) notation inconsistency: you first use “prompt”, then “clue” (ii) listing different scores in the Experiments section feels redundant, as all these scores are already present in the tables; I would rather use this space for more in-depth analysis of the optimization process (iii) some of the Iteration 1 prompts listed in the Appendix (pages 25, 26) look strange: they mention “creating a word problem” rather than solving it, and this is not addressed. I am curious as to what led to the prompts like that: is it an initialization metaprompt artifact?

Some prior works

[1] TextGrad: Automatic "Differentiation" via Text, Yuksekgonul et al.

[2] Trace is the Next AutoDiff: Generative Optimization with Rich Feedback, Execution Traces, and LLMs, Cheng et al.

[3] Large Language Models as Optimizers. Yang et al.

[4] Promptbreeder: Self-Referential Self-Improvement Via Prompt Evolution, Fernando et al.

[5] TRPrompt: Bootstrapping Query-Aware Prompt Optimization from Textual Rewards, Nica et al.

[6] QPO: Query-dependent Prompt Optimization via Multi-Loop Offline Reinforcement Learning, Kong et al.

[7] many other works!

**Questions:**

The questions follow from the drawbacks highlighted in the Weaknesses section

1. Have you compared your method with other SOTA prompt optimization techniques?
2. Have you analyzed the optimization dynamic of different attributes of the agent? Which of them contribute most to the accuracy improvement?
3. What is the motivation behind not using verifiable rewards for the assessment part of the pipeline? For the currently used approach, how do the different components evolve with the iteration number?
4. Have you tried pairing different agents with different LLMs?

---

### Official Review · Reviewer_NkkS · 2025-11-01

**Soundness:** 3
**Presentation:** 3
**Contribution:** 3
**Rating:** 4
**Confidence:** 4

**Summary:**

This paper introduces Agent-GWO, a dynamic prompt optimization framework for large language models (LLMs). The framework leverages collaboration among multiple agents and employs the Grey Wolf Optimizer (GWO) to iteratively refine prompts. During optimization, the top three performing solutions guide the search process for the remaining candidates, enabling efficient and adaptive improvement of prompt quality.

**Strengths:**

- The paper is clearly written and well-structured.
- Extensive evaluation is conducted across LLMs of varying sizes and on both mathematical and hybrid reasoning datasets, demonstrating the framework’s versatility.

**Weaknesses:**

- Comparisons are limited to CoT and CoT-SC, with CoT-SC not clearly defined.
- Broader comparisons with other prompt optimization methods—such as Self-Refine, ReAct, PromptBreeder, PromptAgent, APE, and ProTeGi—are missing.
- The rationale for hyperparameter selection, specifically the values of n and K, is insufficiently explained.
- There is no analysis of latency or computational cost, particularly regarding the impact of increasing the number of tokens.

**Questions:**

- How does the performance of Agent-GWO change when the number of agents n<5? Are all five agents necessary for effective optimization?
- Similarly, how does performance vary when the parameter K<10? What is the impact of smaller values on optimization quality and convergence?

---

### Official Review · Reviewer_fJCL · 2025-11-01

**Soundness:** 3
**Presentation:** 3
**Contribution:** 2
**Rating:** 4
**Confidence:** 3

**Summary:**

Agent-GWO introduces a population-based, dynamic prompt-optimization framework. It models each candidate as an LLM “agent” (a CoT prompt template + sampling hyperparameters). It adapts the Grey Wolf Optimizer (GWO) leader–follower mechanism to iteratively evolve these agents.
In each iteration agents generate Chain-of-Thought traces and answers. An evaluator scores outputs on three dimensions (logical consistency, creativity/ingenuity, and reasoning completeness). The top three agents (α, β, δ) become leaders. The remaining agents’ continuous hyperparameters are updated via leader-centered sampling/weighted averaging. Prompts are adapted by a PROMPTADAPTATION routine (template mixing/keyword imitation). After K iterations the α agent’s configuration is used for inference.
The paper details hyperparameter sampling/clipping, initialization, pseudocode, and an optimization loop. It evaluates Agent-GWO across mathematical and hybrid reasoning benchmarks (GSM8K, MATH, SVAMP, MMLU, BBH, DATE, CLUTRR, etc.) and multiple LLMs (Qwen2.5-Coder-7B, GPT-4.1 variants, GPT-4o-mini, Gemma-3-12b-it). It reports consistent gains over CoT and CoT+Self-Consistency.

**Strengths:**

1. Well-specified algorithm that maps LLM agents to GWO with explicit update rules. The method cleanly instantiates GWO in the LLM setting: agents are parameterized by prompts and decoding hyperparameters; leader–follower updates are given with the standard GWO equations and an averaging rule over α/β/δ; prompts and hyperparameters are adapted iteratively. The formalization reduces ambiguity in “multi-agent” prompt search and makes the procedure implementable and analyzable.

2. Broad and consistent gains across models and benchmarks, with concrete numbers. Across math and hybrid reasoning, GWO consistently outperforms CoT and CoT-SC on GPT-4.1-mini, GPT-4.1-nano, GPT-4o-mini, Gemma-3-12b-it, and Qwen2.5-Coder-7B-Instruct; e.g., GPT-4.1-mini on GSM8K 96.9% vs 88.2% (CoT) and 91.8% (CoT-SC); MMLU 78.3% vs 66.9% (CoT). These results are tabulated clearly.

3. Compatibility with CoT and scaling behavior verified via targeted ablations. The framework composes with CoT and shows further gains when combined; systematic ablations show performance increases with more agents (n) and more iterations, with concrete trends plotted.

4. The paper positions Agent-GWO as an approach that adapts prompts and decoding hyperparameters without modifying model weights, aiming to cut training/inference overhead relative to parameter-tuning baselines.

5. Clear experimental setup, datasets, and models; reasonable steps toward reproducibility

**Weaknesses:**

1.	Evaluation function is under-specified and self-referential: FITNESS/EVALUATION scores combine “logical consistency, creativity, and completeness,” but the paper does not specify the scorer, calibration, prompts for the judge, inter-run variability, or any human validation. If an LLM judge is used without calibration or human audit, improvements can be artifacts of the evaluator rather than real task competence.

2. No quantitative evidence for the low-overhead claim.

3. Claims of broader domains are not backed by corresponding experiments
The abstract cites “social sciences, medical diagnostics, and decision support” as application areas, but empirical sections focus on math and hybrid reasoning benchmarks; no task, metric, or dataset from the named domains is reported.

4. No statistical rigor or error analysis: Results are reported as single accuracies without confidence intervals, standard deviations, or significance tests; qualitative failure analyses are absent.

5. Limited baseline coverage for prompt-optimization/search methods: Core comparisons are to CoT and CoT-SC, plus a mixed appendix table of older/self-training methods; head-to-head against contemporary prompt search/optimization approaches is missing. Without competitive prompt-search baselines, it is hard to attribute gains to GWO mechanics rather than to generic exploration.

**Questions:**

1. The paper mentions medical and social-science decision support as applications. Since the experiments are only on math and reasoning tasks, can you clarify whether the method is intended for those other domains? Can you provide more examples or tone down this?

2. Failure Mode Analysis: Can you provide examples where the method hurts performance or produces low-quality prompts, and how you diagnose such failures?

3. How is the evaluation function instantiated in practice? Specify whether the evaluator is an LLM judge or task accuracy check, the exact evaluation prompts, weighting scheme, and whether multiple evaluation seeds were tested.

---

### Note · Authors · 2025-11-29

I have read and agree with the venue's withdrawal policy on behalf of myself and my co-authors.